# Lineage specific transcription factor waves reprogram neuroblastoma from self-renewal to differentiation

Deblina Banerjee[1,8] ✉, Sukriti Bagchi[1], Zhihui Liu [1], Hsien-Chao Chou [2], Man Xu[1], Ming Sun[1], Sara Aloisi[3,4], Zalman Vaksman[5], Sharon J. Diskin [6], Mark Zimmerman[7], Javed Khan [2], Berkley Gryder [3,8] ✉ & Carol J. Thiele [1] ✉

Temporal regulation of super-enhancer (SE) driven transcription factors (TFs) underlies normal developmental programs. Neuroblastoma (NB) arises from an inability of sympathoadrenal progenitors to exit a self-renewal program and terminally differentiate. To identify SEs driving TF regulators, we use all-trans retinoic acid (ATRA) to induce NB growth arrest and differentiation. Time-course H3K27ac ChIP-seq and RNA-seq reveal ATRA coordinated SE waves. SEs that decrease with ATRA link to stem cell development (*MYCN, GATA3, SOX11*). CRISPR-Cas9 and siRNA verify SOX11 dependency, in vitro and in vivo. Silencing the *SOX11* SE using dCAS9-KRAB decreases *SOX11* mRNA and inhibits cell growth. Other TFs activate in sequential waves at 2, 4 and 8 days of ATRA treatment that regulate neural development (*GATA2* and *SOX4*). Silencing the gained *SOX4* SE using dCAS9-KRAB decreases *SOX4* expression and attenuates ATRA-induced differentiation genes. Our study identifies oncogenic lineage drivers of NB self-renewal and TFs critical for implementing a differentiation program.

Neuroblastoma is a tumor of the peripheral sympathetic nervous system, where despite intensive multimodal therapies, survival is limited to 50% in high-risk cases[1]. Patients with undifferentiated or stroma-poor tumors have a poor prognosis. Tumors overexpress genes regulating cell cycle progression and may have MYCN amplification or copy number alterations (1p36del; unbalanced 11q; 17q+; 7+). MYCN is often amplified via extrachromosomal circular DNA, driving global increases in transcription needed to promote cancer cell proliferation[2,3]. In contrast, low-risk patients have mature, stroma-rich tumors (ganglioneuromas) with a range of differentiated phenotypes and a transcriptome enriched in differentiation genes[4–6].

During development, trunk neural crest cells delaminate from the neural tube and undergo neural differentiation, in which some commit to the sympathoadrenal lineage[7]. Current models indicate NB originates from a failure of neural crest-derived sympathoadrenal progenitor cells to exit a self-renewal program and implement a differentiation program[8]. Cancer cells often inherit their transcriptional profile from their normal cellular counterparts. Their abnormal growth may involve lineage subversion rendering cells unable to respond to normal physiologic cues. At a molecular level, lineage specification and differentiation are dictated by gene regulatory networks controlled by a small number of master transcription factors (MTFs)[9] which are often involved in core regulatory circuits (CRCs).

[1]Cell & Molecular Biology Section, Pediatric Oncology Branch, Center for Cancer Research, National Cancer Institute, Bethesda, MD, USA. [2]Genetics Branch, Center for Cancer Research, National Cancer Institute, Bethesda, MD, USA. [3]Department of Genetics and Genome Sciences, Case Western Reserve University, Cleveland, OH, USA. [4]Department of Pharmacy and Biotechnology, University of Bologna, Bologna 40126, Italy. [5]New York Genome Center, New York, NY 10013, USA. [6]Department of Pediatrics, Division of Oncology, Perelman School of Medicine, Philadelphia, PA, USA. [7]Department of Pediatric Oncology, Dana-Farber Cancer Institute, Boston, MA 02215, USA. [8]These authors contributed equally: Deblina Banerjee, Berkley Gryder. ✉e-mail: banerjee.deblina@gmail.com; berkley.gryder@case.edu; thielec@mail.nih.gov

Lineage-dependency mechanisms often involve lineage-specific master regulatory genes[2,10]. Thus, one current approach to identify critical therapeutic targets is to delineate key transcriptional machinery dependencies[11].

A recent study has shown that several master regulators important in sympathoadrenal neural crest lineages, including PHOX2A/B, GATA2/3, HAND2, SOX4/11, and INSM1[12] form a CRC on which the neuroblastoma tumor cells are dependent for survival and growth[13]. Among them, PHOX2B, GATA3, and HAND2 acts as MTFs in NB in the CRC[13–16]. Thus, like lineage regulating MTFs, they are expressed at high levels and are driven by cell type-specific super-enhancers[11]. In addition, the MYCN oncogene plays a role in aggressive NB, and is assisted by HAND2 (ref. 17). It is expressed throughout the process of migration, survival, and/or differentiation of these migrating neural crest cells, and its levels gradually reduce in differentiating sympathetic neurons[18,19]. However, in neuroblastoma, ectopic expression of MYCN in migrating neural crest cells may induce proliferation and maintenance of neural identity yet limit their differentiation potential when they normally reach the ganglia[18]. This has led to several studies describing the super-enhancer (SE) mediated transcriptional CRCs governing gene expression programs that drive neuroblastoma growth and survival[13–15]. Several lines of investigation indicate that genetic alterations in NB are associated with oncogenic subversion of the transcription factors normally involved in neural crest development[20]. However, it is unclear how this super-enhancer regulatory circuitry is perturbed when tumor cells switch from a cancer cell self-renewal state[21–23] to a differentiated state.

In this study, we carefully characterize the cis-regulatory networks that underpin the reprogramming of neuroblastoma cells for differentiation. We accomplished this by identifying and validating the master regulators driving self-renewal and those necessary to implement a differentiation program in high-risk neuroblastoma tumors.

## Results

### All-trans retinoic acid (ATRA) mediated differentiation leads to dynamic changes in the super-enhancer landscape of MYCN-amplified NB cells

To identify critical SEs driving transcriptional regulators of NB self-renewal and differentiation, we utilized the well-studied model in which ATRA inhibits NB tumor cell growth and induces differentiation[24,25]. A schematic is detailed in Fig. 1a. Consistent with previous results[24], ATRA treatment of KCNR cells led to a 50–90% decrease in cell growth at 4 and 8 days, respectively (Suppl. Fig. S1a). This was accompanied by a 50% decrease in cells in the growth fraction (S + G2/M) of the cell cycle at 8 days (Suppl. Fig. S1b) and morphologic differentiation (Fig. 1b) showing a 4–8-fold increase in neurite length over the time studied (Suppl. Fig. S1c). RNA-seq analysis showed significant downregulation of genes implicated in the NB transcriptional core regulatory complex, such as MYCN, ISL1, and GATA3[13] (Suppl. Fig. S1d, left panel). Gene set enrichment analysis (GSEA) showed a significant downregulation of Benporath Proliferation and MYCN_UP signatures, and a significant upregulation of Frumm differentiation and Cahoy Neuronal signatures in the transcriptome of cells treated with ATRA (Suppl. Fig. S1d, right panel), confirming ATRA's ability to influence these pathways in NB.

To gain insight into the changes in the epigenetic landscape due to ATRA-mediated differentiation, we performed H3K27ac ChIP-seq at indicated time points on KCNR cells, under self-renewing (EtOH-2D) and differentiating states (ATRA 2D, 4D, and 8D). We defined SEs (Fig. 1c) using a modified ROSE algorithm[26], to account for the MYCN amplification under all conditions. In control KCNR cells, SEs spanned a median of 11.9 kb (compared with 1.07 kb for typical enhancers) and had a 12.7-fold higher load of H3K27ac (median intensity for typical enhancers was 980 compared to 12,408 for SEs) (Suppl. Fig. S1e). Total H3K27ac protein levels did not change after ATRA treatment

(Suppl. Fig. S1f). The number of typical enhancers identified in the controls (EtOH-2D) essentially remained constant from 2 days ($n = 9097$) to 4 days ($n = 8872$) and decreased by 20% by 8 days ($n = 7239$). The SEs in controls essentially remained constant over time (data not shown). In the ATRA-treated cells, there was a dynamic reorganization of the enhancer landscape, with the number of enhancers initially decreasing compared to the controls at 2 days but increasing twofold from 2 days ($n = 6661$) to 4 days ($n = 13,698$) followed by an additional 17% increase by 8 days ($n = 16,400$) (Suppl. Fig. S1g). Similarly, the number of SEs in ATRA-treated cells initially decreased compared to the controls but increased 1.5-fold from 2 days ($n = 358$) to 4 days ($n = 593$) (Suppl. Fig. S1h).

Consistent with previously published studies, under self-renewal or control conditions (EtOH-2D), we found MYCN acts as the major driver with the most highly active SE followed by the recently identified components of the NB CRC such as *PHOX2B, HAND2, GATA3*[14,26] (Fig. 1c: EtOH-2D). Upon ATRA treatment, MYCN SE dramatically changed with a significant decrease in H3K27ac levels after 2–4 days (Fig. 1c). In addition to MYCN, SE signals associated with multiple genes such as *SOX11, SOX4, GATA3*, and *GATA2* also changed with ATRA treatment (Fig. 1c).

To understand the dynamic changes in the SE landscape mediated by ATRA, we evaluated the temporal patterns of SE changes and identified 1199 SEs (Supplementary Dataset 1) in KCNR cells under self-renewal and differentiating conditions. All identified SEs were rank-ordered according to their H3K27ac signal. To identify SEs dynamically regulated by ATRA (Fig. 1a), we removed SEs that remained stable over the course of treatment (SD <30%; $n = 143$) or those whose temporal pattern changed under both control and ATRA conditions (Max variance <200 H3K27Ac; $n = 116$). For the remaining SEs ($n = 940$), K-means clustering based on H3K27ac density z-score was used, and identified four temporal patterns of regulated SEs (Suppl. Fig. S1i). The temporal pattern of the H3K27ac Z-score for each cluster is plotted in Fig. 1d–g, and a heatmap of H3K27ac signals of all SEs for the four clusters is shown in Suppl. Fig. S2a. The "Lost" cluster was characterized as a group of SEs ($n = 254$) whose activity either decreased or was completely lost following ATRA treatment (Fig. 1d). The three other SE clusters showed sequential waves of activation in response to ATRA treatment. These were termed the first SE wave cluster at 2D ($n = 174$), the second SE wave cluster at 4D ($n = 355$), and the third SE wave cluster at 8D ($n = 157$) (Fig. 1e–g). Using GREAT to infer the biological output of non-coding regions by analyzing annotations of neighboring genes[27], the Lost SE cluster was found to be enriched in the regulation of signaling pathways and DNA binding followed by stem cell development (Suppl. Fig. S2b). The first wave of gained SEs reflected responses to hormones, biomolecules, and negative regulation of signal transduction and cell communication, while the second and third waves of gained SEs predominantly reflected neural development, cell differentiation, and axonogenesis (Suppl. Fig. S2b) consistent with morphological changes and increases in neurite extensions.

Since neuroblastoma arises due to a block in terminal differentiation, we sought to interrogate the dynamically regulated SEs for the presence of neuroblastoma-associated single-nucleotide polymorphisms (SNPs). Genome-wide association studies (GWAS) of neuroblastoma have identified over a dozen susceptibility loci and these genetic associations have implicated genes such as *CASC15, NBAT1, BARD1, LMO1, DUSP12, DDX4, IL31RA, HSD17B12, HACE1, LIN28B, TP53, RSRC1, MLF1, CPZ, MMP20, KIF15*, and *NBPF23*[28–37]. We intersected NB-associated common SNPs from a recent neuroblastoma GWAS[35] with the dynamically ATRA-regulated SEs in KCNR cells and observed overlap at the *LMO1* locus on chromosome 11p15 (Supplementary Dataset 2). *LMO1* is a transcriptional coregulator that functions as an oncogene in neuroblastoma[16,31]. The most significant SNP, rs2168101 (G > T; *p* value $3.18 \times 10^{-16}$, odds ratio 0.70, 95% CI 0.65–0.76), maps to

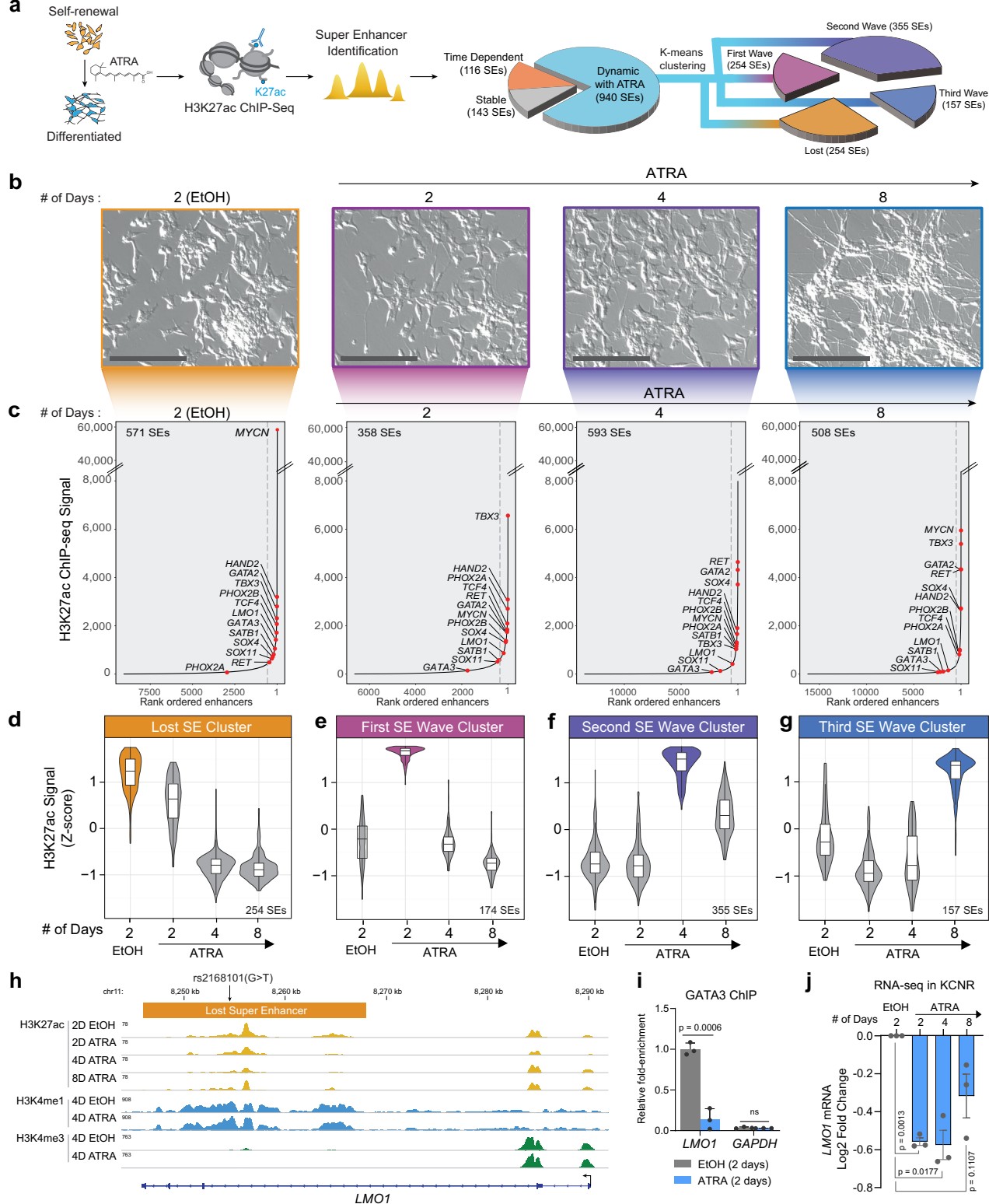

an evolutionarily conserved GATA3 binding site within the LMO1 SE. The risk allele (G) retains the canonical binding site, and this SE drives the expression of LMO1[38]. In contrast, the rs2168101 protective allele (T) ablates GATA3 binding and is associated with low- or no LMO1 expression. The NB cell line KCNR contains the G risk allele associated with tumor LMO1 SE and became a Lost SE cluster candidate. Specifically, ChIP-seq indicated a 70% loss of H3K27ac signal at LMO1 SE after 4 days of ATRA treatment (Fig. 1h), without any change in H3K4me1.

This indicates despite being still present in the cells the SE has reduced activity. In addition, the promoter signal of LMO1 also remained unchanged, as observed by constant H3K4me3 levels. Thus, any change in downstream gene expression is likely due to the loss of this SE. Since the risk allele retains a conserved GATA3 motif, we used ChIP analysis to examine GATA3 binding at this locus and found a 75% decrease in GATA3 binding in this region after treatment with ATRA (Fig. 1i). This was accompanied by a 1.46-fold decrease in LMO1 mRNA

**Fig. 1 | ATRA treatment of NB cells reveals a dynamic change in the super-enhancer (SE) landscape. a** Schematic representation of the study design and step-by-step approach to identify SEs and downstream clusters after ATRA treatment. SEs were divided into subgroups: Stable with ATRA: SD <30% over 2, 4, and 8 days of ATRA treatment; Time Dependent: - Max variance <200 H3K27Ac signal in control (EtOH) and ATRA conditions. The remaining SEs were defined as Dynamic with ATRA and subjected to K-means clustering to identify temporal pattern. Four clusters of SEs were identified. **b** Images of NB cells showing the time-dependent effect of 5 μM ATRA treatment on cell morphology, representative of biological triplicate experiments. Scale bars are 200 μM. **c** H3K27ac binding at super-enhancers ranked by increasing signal. SEs were identified beyond the inflection point of increasing H3K27ac load (indicated by a gray dashed line). Select SE target genes are highlighted. The SEs were linked to putative target genes by proximity to expressed genes. **d**–**g** Violin plot showing the four clusters of SEs identified by z-score normalized H3K27ac signal. The bar on the right shows the number of SEs in that cluster. **d** A group of $n = 254$ SEs is highly active in the control or self-renewing cells and is gradually lost in the ATRA-treated differentiated cells. **e** A group of

$n = 174$ SEs is specifically activated 2 Days after ATRA treatment. **f** A group of $n = 355$ SEs is specifically activated 4 Days after ATRA treatment. **g** A group of $n = 157$ SEs is specifically activated 8 Days after ATRA treatment. Box plots show the median with quartiles, whiskers show the 1.5 × interquartile range in the data, are from a single ChIP-seq experiment per time point, and are overlaid with the distribution shown as a violin plot. **h** Representative ChIP-Seq tracks for H3K27ac, H3K4me1, and H3K4me3 in EtOH and ATRA-treated cells in KCNR cells, showing loss of LMO1 SE after ATRA treatment harboring the protective rs2168101(G > T) SNP. **i** Bar graph of GATA3 ChIP-qPCR showing a relative decrease in GATA3 binding at specific LMO1 SE region following 4 days of ATRA treatment. $P$ values were calculated with an unpaired, two-tailed $t$-test. Bars show the mean, with error bars representing the standard deviation. Source data are provided as a Source Data file. **j** Bar graph showing a relative decrease in LMO1 mRNA expression. $P$ values were calculated with a paired two-sided student's $t$-test. Bars show the mean, with error bars representing the standard error of measurement. Source data are provided as a Source Data file.

by RNA-seq (Fig. 1j). Taken, together, this suggests a potential involvement of the lost cluster SEs in driving NB tumorigenesis.

## Super-enhancers fluctuate in sequential waves

In each cluster, proximity analysis within topological domains was used to associate SEs with their respective genes (Fig. 1d), and Ingenuity Pathway Analysis (IPA) revealed the predicted cellular localization of the genes. Some 31–37% of SE-driven genes in the lost cluster and gained first-wave cluster encoded proteins that localized to the nucleus, while in the gained second and third-wave clusters, SEs associated with nuclear genes decreased. In these clusters 40% of the SEs regulate genes whose protein products associate with cytoplasmic processes (Suppl. Fig. S3a). Of the SE-associated genes whose proteins localized to the nucleus, the greatest fraction encoded transcriptional regulators (Suppl. Fig. S3b).

To identify the transcriptional regulators driving NB self-renewal and those contributing to the differentiated cell phenotype, we focused on the analysis of SEs that are associated with transcription factors (TFs) (Suppl. Fig. S3c). SEs driving TFs made up 12% (31/254) of SEs in the lost cluster, while they comprised 9.7% (17/174) of the gained first wave, 13.8% (49/355) of the gained second wave, and 9.5% (15/157) of the gained third wave of SEs. RNA-seq indicated that changes in SEs signal (Suppl. Fig. S3d) corroborated changes in the downstream gene expression (Suppl. Fig. S3e). In the lost cluster, the mean expression of TF mRNAs significantly decreased over the course of the 8 days of ATRA (Suppl. Fig. S3e). Similarly, the mean expression of TF mRNAs associated with SEs in the gained first and second waves was significantly increased compared to their expression in untreated cells (Suppl. Fig. S3e). Since there were no significant differences in the TFs in the gained 3rd wave cluster over the course of ATRA treatment, this group was not included in further analyses.

To focus on master regulators driving NB self-renewal and those contributing to the differentiated cell phenotype, we filtered the analysis for SEs associated with transcription factors (TFs) whose temporal pattern of mRNA expression showed a direct relationship with its linked SE signal, as measured by H3K27ac load at the SE. Since SEs regulate the expression of a downstream gene, we focused on genes whose loss or gain of SEs correlated with a decrease or increase in gene expression ($r > 0.45$) (Fig. 2a and Supplementary Dataset 3). The decreased SE signal in the lost cluster was associated with a significant decrease in the expression of their associated TFs by 8 days ($p < 0.001$) (Fig. 2b, c). There were significant increases in the expression of TFs associated with the first and second gained SEs at 2–4 days (Fig. 2b, c). Heatmaps showed changes in the gene expression of TFs in each cluster (Fig. 2d). The changes in SEs and gene expression were validated in LAN5, another MYCN-amplified NB cell line responsive to ATRA. Over 70% of the TFs examined in

Fig. 2d showed similar changes in LAN5 at the expression level (Fig. 2e) (Suppl. Fig. S4a–c).

The lost cluster included SEs driving TFs such as MYCN, GATA3, and TBX2 that have been identified as a part of CRC controlling cell growth and proliferation in NB[13,14,26,39]. In addition, this cluster included the SOX11 TF, which is involved in the proliferation of sympathetic ganglia at early stages during normal sympathetic nervous system development[40]. Using HOMER Motif finder, we found that in sites of decreased H3K27ac binding (focusing on nucleosome-free regions between H3K27ac peaks), the motifs in lost sites were enriched in the ATOH1/NeuroD/ASCL1 bHLH (motif: CAGCTG), the GATA Zf motif (GATAAG), and the LHX2/ISL1 homeobox TAATT(A/G). We observed a loss in the SE signal associated with MYCN, SOX11, and GATA3 in both KCNR and LAN5 cells (Fig. 2f). Importantly, expression of these TFs (with the exception of TWIST1) also decreased at the protein level in KCNR cells (Suppl. Fig. S3f).

The first and second SE wave clusters included SEs linked to genes such as *SOX4, PPRAG, TBX3, ETS1, HEY1, GATA2,* and *PHOX2A* with increased mRNA expression (Fig. 2d, f). Among them, SOX4, GATA2, and TBX3 also showed an increase at the protein level (Suppl. Fig. S3f), while ETS1 exhibited protein level decreases indicative of post-translational regulation. Figure 2f shows changes in SEs driving SOX4 and GATA2 in KCNR and LAN5 cells. GATA2 is a potent inhibitor of the proliferation of neuronal progenitors and indispensable for the differentiation of several tissues during embryogenesis[41] and SOX4 expression is increased in the late stages of sympathetic nervous system development[40].

To assess the function of the genes in the various clusters, we utilized the DepMap portal to mine the CRISPR 21Q3 Public in the 31 NB cell lines contained within the 1032 cancer cell dataset. This analysis showed enrichment in dependency genes among the 29 TFs driven by SEs in the lost SE cluster (Suppl. Fig. S3g), consistent with a role for these genes in driving self-renewal. Overall, the mean dependency score of the genes driven by dynamically regulated ATRA gained first, second, and third SE waves did not show evidence for dependence (Suppl. Fig. S3g). We validated this in LAN5 cells (Suppl. Fig. S4). This suggests that genes with gained SEs after ATRA treatment are not essential for self-renewal or proliferation but rather would be critical to mediate differentiated properties of NB cells.

## SOX family-driven regulatory circuits correlate with neuroblastoma prognosis

SEs frequently regulate the expression of MTFs of particular cell lineages and form a CRC, in which the binding motifs for each of the cell's MTFs are found in the SEs of that cell type[42]. Because we identified changes in the SE landscape as a result of ATRA treatment, we wanted to investigate if these changes also affected the

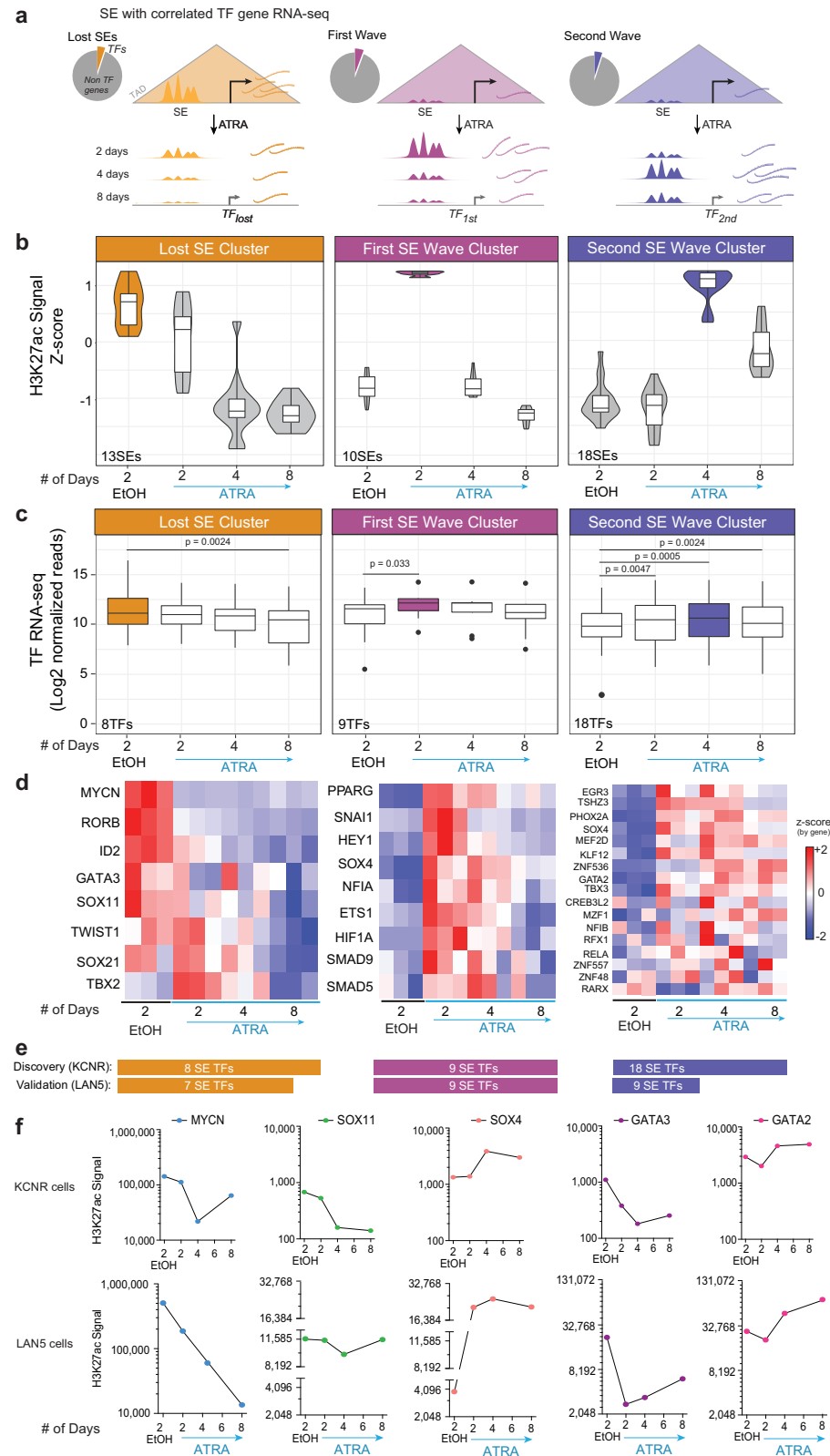

composition of the CRC. We linked each CRC factor to the SEs activating each other factor, and visualized this using linked enhancer-activated factors (LEAF) plots. Each node (Fig. 3a) represents a transcription factor, and each green arrow indicates the presence of a given transcription factor's DNA binding motif in the active SE regulating the TF to which the arrow points. The size and order of the nodes are proportional to the total mRNA expression of each TF.

We found that after ATRA treatment, the SOX11 factor is lost from the CRC owing to its lost SEs (SE#1 near the gene, and SEs #2 and #3 downstream intergenic). Interestingly, SOX4 rose to the top of the CRC factors in the presence of ATRA (Fig. 3a). The inward binding of all TFs in the SOX4/11 SEs, inferred from the presence of a given TF's motif in a SE, decreased for SOX11 and increased for SOX4 upon ATRA treatment (Fig. 3b). The outward binding of SOX4 increased (Fig. 3b) as the

**Fig. 2 | SE-driven transcription factors drive the proliferation and differentiation of neuroblastoma cells. a** A schematic representation of the analysis steps following SEs identification in each cluster. As TFs drive the core regulatory circuitry of a cell, analysis was further restricted to SEs linked only to TFs (Pie charts), and their expression was evaluated at the mRNA level. **b** Violin plot showing the four clusters of SEs driving TFs ($r^2 \geq 0.45$) normalized by z-score of H3K27ac signal. The lost cluster consists of 13 SEs, the first wave cluster has 10 SEs, and the second wave cluster consists of 18 SEs. Data were from a single ChIP-seq experiment per time point. Box plots overlapping the violin plot depict the center line = median, box bounds = quartiles, whiskers = 1.5*interquartile range. **c** Box plots showing expression of the TFs driven by the SEs in each cluster. The 13 SEs in the lost cluster drives eight TFs where the loss in SE signal leads to a significant decrease in the expression of the corresponding TFs after 8 days of ATRA treatment. Similarly, gain in 10 SEs in the first wave cluster significantly increased expression of its downstream 9 TFs after 2 days and 4 days of ATRA treatment. And 18 SEs gained in the second wave led to significant gain in the expression of the downstream 18 TFs. *P* values were calculated with a paired, two-tailed *t*-test. Box plots depict the center line = median, box bounds = quartiles, and whiskers = 1.5*interquartile range. **d** Heatmap showing expression of individual TFs in each cluster in controls and after 2, 4, and 8 days of ATRA treatment in three biological replicates for each condition. Source data are provided as a Source Data file. **e** Number of SE-regulated TFs discovered in KCNR and validated in LAN5. **f** Line graph showing dynamic changes at specific SE with ATRA treatment in KCNR and LAN5. H3K27ac signal at the SE associated with MYCN, SOX11, and GATA3 is decreased, whereas the signal at SOX4 and GATA2 are increased. ChIP-seq signal is normalized as reads per million mapped reads (RPM) summed over the entire SE region.

abundance of the SOX4 motifs increased in SEs associated with NB cell differentiation. Seven TFs that gained *SOX4* binding also had increased mRNA expression after ATRA treatment (Fig. 3c).

Phenotypic variants with adrenergic (ADRN) or mesenchymal (MES) biologic characteristics have been identified in NB cells[15]. KCNR cells are enriched in ADRN gene signature and under both control and ATRA treatments, cluster with other NB cells with an ADRN phenotype (Suppl. Fig. S5a, b). In the lost SE cluster, the SE predicted to drive *SOX11* expression had elevated levels of H3K27ac in multiple other NB cell lines, whereas minimal H3K27ac was seen at the *SOX4* SE (Fig. 3d).

SOX11 and SOX4 are members of the SOX-C family of transcription factors[43]. As sympathoadrenal neural cells differentiate, early expression of Sox11 is followed by increasing expression of Sox4 as Sox11 levels decline[40]. Since little is known of the function of SOX genes in NB, we evaluated the expression of SOX11 and SOX4 in a cohort of 395 normal and tumor tissues sequenced in the TARGET dataset. Relative to the expression in normal tissues, we observed higher expression of SOX11 and SOX4 in NB tumor tissues (Suppl. Fig. S6a). In primary NB tumors (Dataset: SEQC), high levels of SOX11 were associated with a poor outcome (Fig. 3e left panel; Bonferroni *p* = 1.7e-11), whereas elevated levels of SOX4 were associated with a better prognosis (Fig. 3e right panel Bonferroni p = 5.8e-18). The prognostic significance of SOX11 and SOX4 in primary tumors was replicated in an additional R2 database (Kocak dataset, Suppl. Fig. S6b), even when only patients with Stage 3, 4 are evaluated (Suppl. Fig. S6c). Assessment of primary NB patients stratified for risk (database: R2 SEQC; *n* = 498 tumors) reveals increased expression of SOX11 in high-risk cases (Mann–Whitney test, *p* < 0.0001), whereas higher expression of SOX4 was found in low-risk cases (Mann–Whitney test, *p* < 0.0001) (Fig. 3f).

### Gained cluster super-enhancer contributes to increased *SOX4* mRNA

Having observed the gain of the *SOX4* SE and increases in *SOX4* expression with ATRA, we performed Hi-C in KCNR cells to evaluate the prediction of *cis*-interactions between the SEs and the *SOX4* promoter and showed that they occur within the same TAD boundaries (Fig. 4a). We evaluated changes in the chromatin accessibility in this region by ATAC-seq and did observe chromatin opening in the newly gained SE region after ATRA treatment (Fig. 4b). The SE linked to *SOX4* gained H3K27ac signal after ATRA treatment (Fig. 4b) was associated with a 3-fold increase in mRNA (Fig. 4c-top panel). Similar observations were made in the LAN5 NB cell line where ATRA treatment led to a gain of the *SOX4* SE (Fig. 4b) that was associated with a threefold increase in *SOX4* mRNA (Fig. 4c-bottom panel).

To assess the functional contribution of this SE to *SOX4* mRNA expression, we targeted the SE with two sgRNAs, using the dCas9-KRAB one-vector system. The KRAB domain leads to reversible silencing of the chromatin by deposition of H3K9me3 modifications at the loci targeted by the dCas9[44]. We evaluated changes in *SOX4* expression after 2 days of ATRA treatment (Fig. 4d, left panel) and

observed that the sgRNAs led to a 2.5 to 13-fold increase in H3K9me3 signal over the negative control at the specific sgRNA loci compared to minimal or no enrichment for the empty vector (Fig. 4d, right panel). This was accompanied by a significant decrease (*p* < 0.001) in the ATRA-mediated upregulation of *SOX4* expression (Fig. 4e), demonstrating that the physically linked *SOX4* SE regulates *SOX4* mRNA expression. No changes in *SOX4* mRNA levels were observed in the control cells between negative sgRNA and SE sgRNA transduced cells (Fig. 4e, left panel), which could be due to the extremely low basal level of H3K27ac signal at this region. The decrease in sgRNA-mediated SOX4 expression was accompanied by significant decreases (*p* < 0.001) in ATRA-mediated upregulation of *DPYSL3* and *TUBB3* mRNA levels (canonical differentiation markers) in these cells (Fig. 4e, middle and right panel).

### SOX4 enhances ATRA-mediated differentiation of KCNR cells

We next performed SOX4 Cut and Run analyses in KCNR cells under control or self-renewing and after 4 days of ATRA treatment. ATRA treatment showed a significant increase in genome-wide binding of SOX4 at 1041 sites (Fig. 5a). These new SOX4 binding sites were strongly enriched in the SOX4 motif (CTTTGTTCC, *p* = 1e-242) (Fig. 5b and Supplementary Dataset 4) and were found near genes which on average had increased transcriptional output (Fig. 5c and Supplementary Dataset 5). Gene ontology enrichment analysis revealed an enrichment in GO biological terms, including negative regulation of transferase activity, histone ubiquitination, glia cell migration, and noradrenergic neuron differentiation (Fig. 5d).

The finding of increases in SOX4 peaks in the promoter regions of *DPYSL3* and *TUBB3* after ATRA treatment (Fig. 5e) supports a role for SOX4 in contributing to ATRA-mediated differentiation. To further evaluate the role of SOX4 in NB differentiation, we generated doxycycline-inducible SOX4 gain of function KCNR cells (Suppl. Fig. S7a). Increases of SOX4 had no significant effect on KCNR cell growth (Suppl. Fig. S7b), nor evidence of morphologic differentiation (data not shown) or increases in expression of differentiation genes (Suppl. Fig. S7c). To evaluate the role of SOX4 in retinoid-induced differentiation, we generated doxycycline-inducible KCNR-shSOX4 cells. Doxycycline treatment led to at least a 50% decrease in *SOX4* mRNA (Fig. 5f) and protein levels (Fig. 5g). Inhibition of SOX4 expression attenuated the ability of ATRA to induce differentiation markers DPYSL3 (Fig. 5h, left panel) and TUBB3 (Fig. 5h, right panel) and resulted in a 20% decrease in ATRA-induced neurite extension (Fig. 5i). This rescue was confirmed transcriptome-wide (Fig. 5j) by GSEA for SOX4 activated genes and mesenchymal signature genes. These data indicate that SOX4 acts to enhance ATRA-mediated differentiation in neuroblastoma.

### Lost cluster super-enhancer drives SOX11 and contributes to NB self-renewal

As we observed the gain of SOX4 coincided with the loss of SOX11 from its dominant position in the CRC during ATRA-induced differentiation,

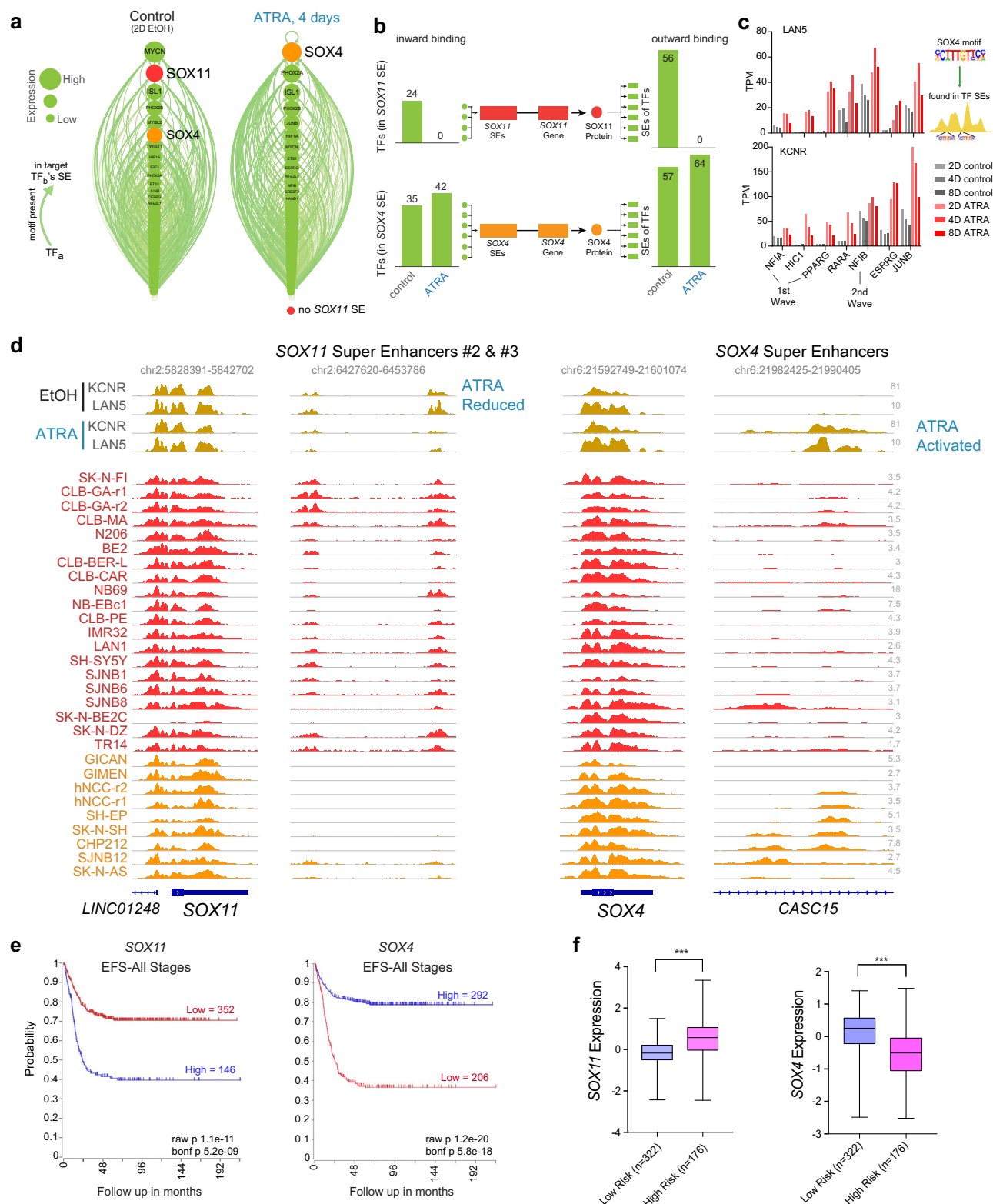

**e**

SOX11
EFS-All Stages

Low = 352
High = 146

raw p 1.1e-11
bonf p 5.2e-09

SOX4
EFS-All Stages

High = 292
Low = 206

raw p 1.2e-20
bonf p 5.8e-18

**f**

we evaluated if the SEs distal to *SOX11* were indeed driving it directly. The SOX11 linked SE was lost after ATRA treatment (Suppl. Fig. S8a) with an 80% decrease in H3K27ac signal and was accompanied by a significant decrease in SOX11 mRNA (*p* < 0.001) (Suppl. Fig. S8b) and protein levels (Suppl. Fig. S3f). To evaluate *cis*-interactions between the SEs and the *SOX11* promoter, we performed Hi-C in KCNR cells and observed *SOX11* and its SEs are encircled within the same TAD boundaries (Fig. 6a). This provided direct evidence linking the

bioinformatically-associated *SOX11* SE to the *SOX11* TSS. To assess the functional contribution of this SE to *SOX11* mRNA expression, we targeted the SE with three sgRNAs (Fig. 6b, left panel) using the dCas9-KRAB system. In two of three sgRNAs, we observed a 50–80-fold enrichment in the H3K9me3 signal over negative control at specific sgRNA loci, compared to no enrichment for the empty vector at the targeted *SOX11* SE (Fig. 6b, right panel). This was accompanied by a 1.5–2-fold decrease in H3K27ac at the same locations (Fig. 6c).

**Fig. 3 | Switching of lineage-specific transcription factors induces differentiation in NB cells. a** LEAF (Linked Enhancer-Activated Factors) plots of CRC linked by motif presence in their active enhancers, with nodes ranked by mRNA expression in descending order, in controls (EtOH treated), and 4 days of ATRA treatment. **b** Inward binding of NB TFs in SOX4 and SOX11 SEs (left) and outward binding in all TF proximal SEs (right) before and after ATRA treatment. Data were from day 4 in KCNR cells and is similar to connectivity at day 2, and is similar to observed changes in LAN5 cells. **c** Bar graph showing an increase in expression of the seven transcription factors that gain SOX4 inward binding after ATRA treatment. **d** ChIP-seq track of H3K27ac in ATRA-treated KCNR and LAN5 cells (top, gold), compared to

group state epigenomics of H3K27ac in other Adrenergic NB cell lines and primary tumors (middle, red) and Mesenchymal NB cell lines and tumors (bottom, orange) at both SOX11 with one of its SEs, and SOX4 with its SE. **e** Kaplan–Meier plots based on the expression of SOX11 (left panel) and SOX4 (right panel) in tumors from NB patients at all stages (R2 database: SEQC dataset). $P$ values were calculated with a log-rank test, with Bonferroni correction. **f** RNA expression of *SOX11* and *SOX4* among low and high-risk NB patients. ***$P < 0.0001$, two-sided students $t$-test with Welch's correction. Data from R2 database: SEQC dataset. Box plots show a median with quartiles, and whiskers show the 1.5 × interquartile range.

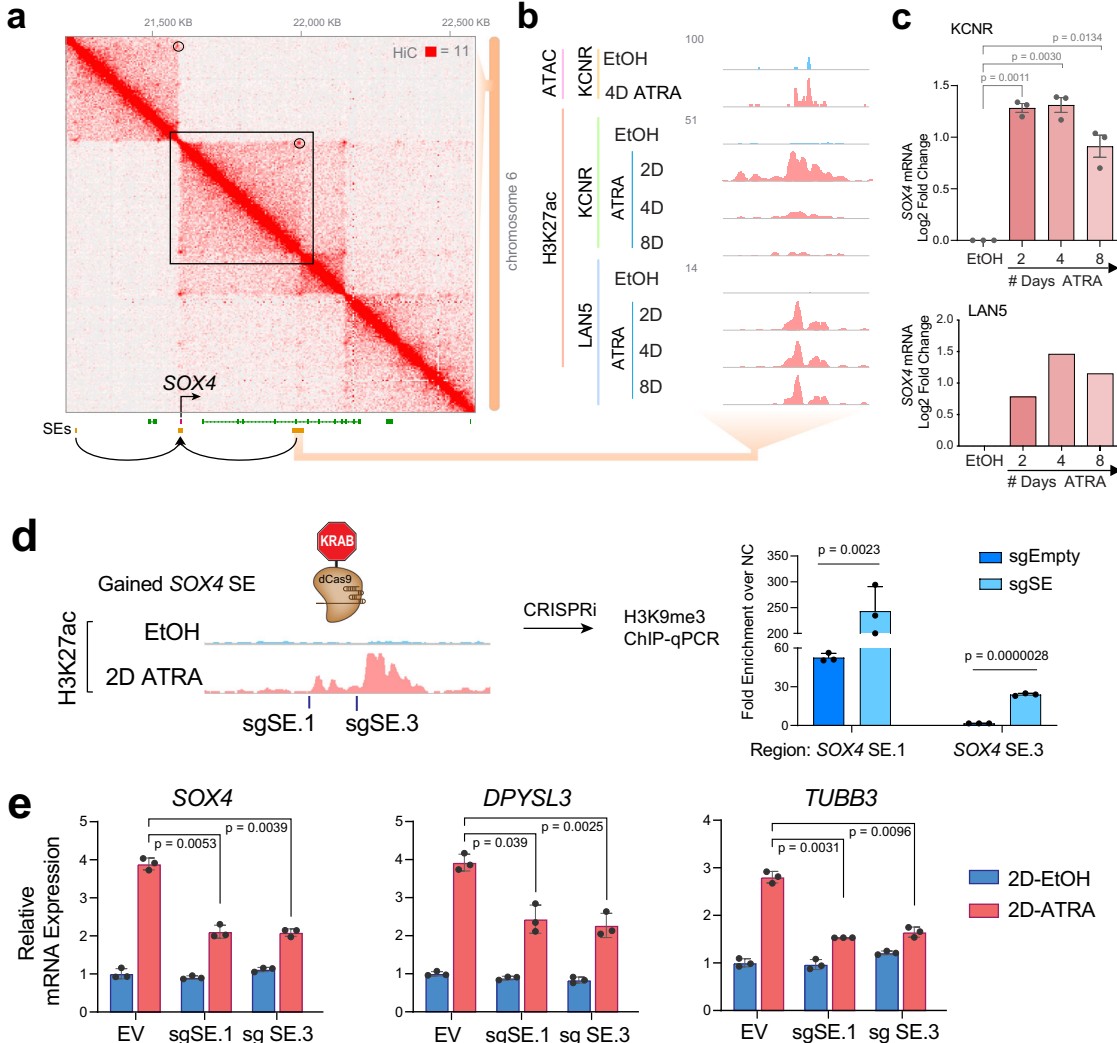

**Fig. 4 | SOX4 silencing leads to inhibition of RA-mediated differentiation of KCNR cells. a** Hi-C contact profile surrounding SOX4 locus in KCNR cells with gained SEs shown in both adjacent TADs. **b** Zoom in on the SE gained, intronic to CASC15. Representative ATAC-seq and ChIP-Seq tracks for H3K27ac in control (EtOH: blue) and ATRA-treated cells (pink), showing an increase in H3K27ac peaks at SOX4 SE in KCNR and LAN5 cells. **c** Bar graph showing relative mRNA levels (as normalized reads) of SOX4 after ATRA treatment in KCNR and LAN5 cells. A log$_2$-fold change increase in SOX4 expression was observed post-ATRA treatment. In KCNR, data were means ± SD for $n = 3$ replicates; $P$ values were generated from paired $t$-test with Welch's correction. **d** Guided suppression of the gained SOX4 SE. Bar graph of H3K9me3 ChIP-qPCR showing increased H3K9me3 at specific gained

SOX4 SE regions following CRISPR-dCas9-KRAB targeting of SOX4 SE. sgRNAs targeting SOX4 SEs were compared to empty vector (sgEmpty) and were performed in KCNR cells. $P$ values were generated from paired $t$-test with Welch's correction across $n = 3$ replicates. Bars show the mean, with error bars showing the standard deviation. Source data are provided as a Source Data file. **e** Bar graph showing suppressed SOX4 (left panel), DPYSL3 (middle panel) and TUBB3 (right panel) mRNA induction following CRISPR-dCas9-KRAB targeting of SOX4 SE. sgRNAs targeting SOX4 SEs were introduced into KCNR cells treated with control or ATRA for 2 days. EV empty vector. $P$ values were generated from paired $t$-test with Welch's correction across $n = 3$ replicates. Bars show the mean with error bars showing SEM. Source data are provided as a Source Data file.

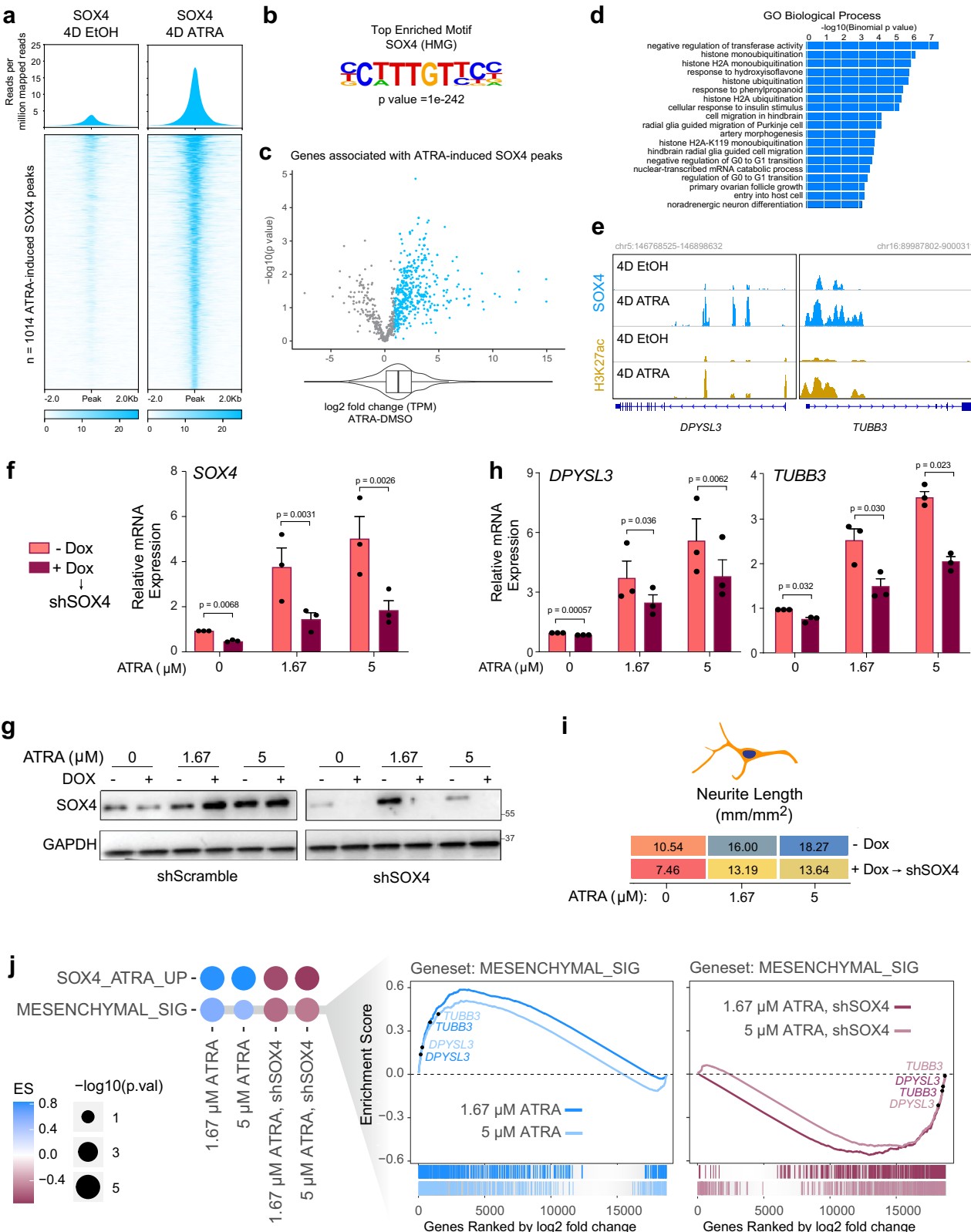

**f** — **-Dox / +Dox → shSOX4**

**i** Neurite Length (mm/mm²)

| | | | |
|---|---|---|---|
| 10.54 | 16.00 | 18.27 | - Dox |
| 7.46 | 13.19 | 13.64 | + Dox → shSOX4 |

ATRA (µM): 0    1.67    5

Targeted chromatin modification swapping at the *SOX11* SE resulted in a significant decrease in *SOX11* expression (*p* < 0.001), and repression of any single SE was insufficient to completely shut down *SOX11* expression indicating an additive effect from these elements on total gene expression output (Fig. 6d). These results functionally confirmed this lost SE directly regulates *SOX11*.

To evaluate the biological role of SOX11 in Neuroblastoma, we further performed SOX11 CUT&RUN under self-renewing and 4 days ATRA-treated KCNR cells. Our analysis showed a small percentage of genes (*n* = 182) lost SOX11 binding (Fig. 6e), and their expression were decreased after ATRA treatment in both KCNR and LAN5 cells (Fig. 6f and Supplementary Dataset 6).

**Fig. 5 | ATRA-mediated differentiation leads to enhanced SOX4 binding on differentiation genes. a** SOX4 Cut and Run analyses showing an increase in SOX4 genome-wide binding after 4 days of ATRA treatment. Upper panel: Composite plots showing SOX4 signal intensities (reads per million mapped reads) at SOX4 high confidence peaks ($n = 1014$) in 4D EtOH (left) and 4D ATRA (right) treated cells. Lower panel: Heatmaps of SOX4 peak intensity at SOX4 high confidence peaks ($n = 1014$). Each row represents a genomic location and is centered around SOX4 peaks, extended 2 kb in each direction, and sorted by SOX4 signal strength. **b** Hypergeometric Optimization of Motif Enrichment (HOMER) analysis identified SOX4 binding motifs at SOX4 peaks using the HOMER package (homer.salk.edu/homer/ngs/peakMotifs.html). *p* statistic is calculated using the HOMER statistical comparison against size-matched DNA sequences from randomly selected background genomic sequences. **c** Volcano plot of log2 fold change (L2FC) of genes associated with ATRA-induced SOX4 peaks in KCNR cells. The box plot underneath and the overlapping violin plot depict the center line = median, box bounds = quartiles, and whiskers = 1.5*interquartile range. *P* values were calculated using the Wald test with Deseq2 across three biologically independent experiments. **d** Gene ontology terms associated with genes found using GREAT analysis

(great.stanford.edu) on SOX4 constituent peaks. **e** Tracks showing an increase in SOX4 (blue) and H3K27ac (yellow) peaks after ATRA treatment at DPYSL3 and TUBB3 locus in KCNR cells. **f** qRT-PCR analysis showing relative mRNA levels of SOX4 (left) after doxycycline-induced inhibition. 50% inhibition of SOX4 mRNA was obtained after doxycycline treatment. The ATRA-mediated increase in SOX4 levels were also reduced ~60–70% on doxycycline treatment. *P* value were calculated using a paired ratio *t*-test across $n = 3$ replicates. Bars show the mean, error bars represent the standard error of measurement. **g** Western blots showing DOX-induced downregulation of SOX4 (via shRNA SOX4, right) during ATRA-induced expression of SOX4 (with shRNA scramble control on the left). Data were representative of $n = 2$ biological replicates. **h** SOX4 reduction slowed ATRA-induced activation of differentiation marker genes *DPYSL3* (left panel) and *TUBB3* (right panel). *P* value were calculated using a paired ratio *t*-test across $n = 3$ replicates. Bars show the mean, error bars represent the standard error of measurement. **i** Heatmap showing differentiation index as measured by neurite length. SOX4 silencing reduced ATRA-mediated differentiation of KCNR cells. **j** GSEA showing inhibition of ATRA-mediated mesenchymal signature upon silencing of SOX4.

## SOX11 inhibition decreases cell growth and tumor volume in NB cells

Since SOX11 is highly expressed in NB cells and tumors, we evaluated whether NB is dependent on SOX11 for self-renewal. We first analyzed the genome-wide CRISPR library screen of 391 cancer cell lines[13] and found that among various tumor types (Supplementary Dataset 9), NB cells are the most selectively dependent on SOX11 compared to all cell lines analyzed (Fig. 7a). However, among different NB cell lines there are a range of dependencies (Suppl. Fig. S8c and Supplementary Dataset 7) with 9/32 NB cell lines showing dependency (range = −1.2 to −0.6). In KCNR cells, the siRNA inhibition of SOX11 caused a 90–100% inhibition of SOX11 mRNA and protein levels (Fig. 7b) and a significant decrease in relative cell number (Suppl. Fig. S8d). Inhibition of SOX11 also resulted in significant decreases in the relative growth, as measured by confluence, in four of six additional NB cell lines tested (Fig. 7c).

To gain further insight into the potential mechanisms contributing to SOX11-mediated growth inhibition and differentiation in NB cells, we evaluated genome-wide changes in the transcriptome by RNA-seq (Suppl. Fig. 8e). We found that the genes regulated by SOX11 were involved in nervous system growth and development (Suppl. Fig. 8f, left panel) and cellular growth and proliferation (*p* value 8.05E-06 – 1.21E-23) (Suppl. Fig. 8f, right panel) amongst other molecular functions (Supplementary Dataset 8). Knockdown of SOX11 significantly increased NTRK1 mRNA levels (Suppl. Fig. S8g, upper panel), and elevated levels of NTRK1 are found in the tumors of NB patients that have a good prognosis[45,46]. Inhibition of SOX11 caused an increase in the differentiation-associated phenotype of neurite extension (Suppl. Fig. S8g, lower panel), an effect that was further amplified by NGF.

To further evaluate the role of SOX11 in in vivo models, we generated two shSOX11 RNAi cell lines. In these cell lines, there was a 90% inhibition of SOX11 protein levels (Fig. 7d, upper panel) that was associated with an 80% decrease in cell growth after 96 h of in vitro culture (Fig. 7d, lower panel). Orthotopic implantation of control shRNA and shSOX11 KCNR cells showed a significant decrease ($p < 0.0001$) in tumor weight for shSOX11 RNA KCNR cells compared to shControl RNA KCNR cells (Fig. 7e).

Cancer cells have a remarkable dependency on the transcription factors that form an autoregulatory CRC, where each member directly regulates the expression of its own gene as well as those encoding the other CRC transcription factors[13–15]. To determine the impact of SOX11 on other components of the CRC, we evaluated the expression of the other CRC TFs after SOX11 knockdown. Indeed, decreased SOX11 expression led to decrease in mRNA levels of CRC targets like *MYCN, PHOX2B, ISL1*, and *TWIST1* in KCNR cells (Fig. 7f). This observation was

further supported by decreased protein levels of CRC targets MYCN, PHOX2B, ISL1 and TWIST1 in shSOX11 cell lines (Fig. 7g). However, silencing of NB CRC components GATA3, PHOX2B, ISL2, and TBX2 (Suppl. Fig. S8h, upper panel) did not lead to decreases in SOX11 (Suppl. Fig. S8h, lower panel) suggesting that SOX11 may be so well supported by TF binding at its SE that loss of any one of these CRCs individually was not detrimental to its expression. In fact, silencing of either MYCN or HAND2 led to increases in SOX11, suggesting complex wiring of these TFs. Loss of multiple CRC members at multiple SEs is likely needed to stop SE activation, and dCas9-KRAB inactivation at any one SE constituent was sufficient to downregulate the *SOX11* partially (Fig. 6d). That silencing of SOX11 in both siSOX11 and shSOX11 lines leads to decreases in CRC target expression including MYCN while the silencing of MYCN leads to increases in SOX11 suggests that SOX11 may function upstream of MYCN which exerts feedback inhibition on SOX11 expression. This idea agrees with recent evidence that CRC interactions are not always feed-forward for each TF and each SE[47]. SOX11 and SOX4 bind to these locations (Suppl. Fig. S9a, b). Our data taken together, suggests a model in which SOX11 sits atop the CRC hierarchy in self-renewing NB, and its displacement in some NB cell lines is important for rewiring to a differentiated state. This model (Fig. 7h) illuminates the cell-identity-determining circuits responsible for giving retinoic acid such potent antitumor and pro-differentiation capacity in the treatment of NB patients.

## Discussion
In this study, we identify specific temporal and dynamic spatial regulatory events involving SE-driven genes that propel neuroblastoma cells from a cancer cell self-renewal/proliferative state to a differentiated state. We find there is an initial loss of a cluster of SEs driving processes associated with stem cell development, DNA binding, and kinase activity followed by the sequential gain of processes regulating signaling events that ultimately lead to the gain of SEs driving processes regulating neural differentiation. By focusing on SEs driving TFs, we find the cluster of lost SEs drives many of the sympathoadrenal lineage specifying TFs integral to the NB CRC[14,16,39,48]. However, we also identify SEs regulating *SOX11* and *SOX4* expression that are coordinately lost and gained during the differentiation process. Similar to MYCN, the loss of the SE driving *SOX11* expression decreases NB cell growth, making NB cells more responsive to differentiation signals. The gain of SE increases SOX4 expression and is associated with the enhanced differentiation state, as functional studies showed decreases in *SOX4* expression attenuate ATRA-mediated differentiation.

Our study uncovers an important role for SOX11 and SOX4 in NB biology. SOX genes encode TFs that bind DNA through conserved high-mobility group domains and have been shown to function as

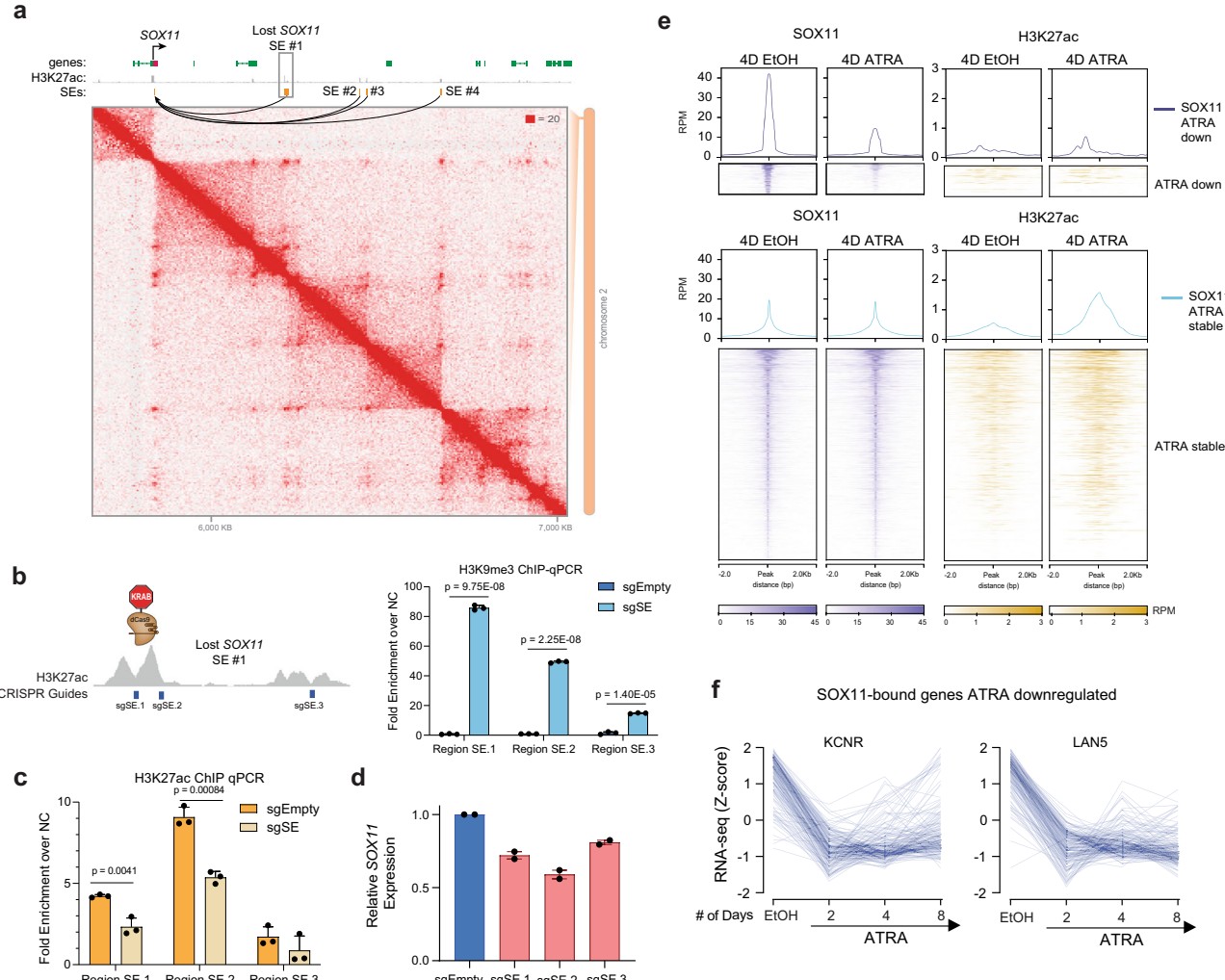

**Fig. 6 | SOX11 expression is regulated by SEs in NB cells. a** Hi-C contact map surrounding SOX11 locus in KCNR cells. Two lost SEs connected to the SOX11 TSS are highlighted, with the SOX11 gene and the largest lost SE (#1) labeled. **b** Left panel: Guided suppression of the lost SOX11 SE. Right panel: Bar graph of H3K9me3 ChIP-qPCR showing the mean relative increase in H3K9me3 peaks at specific SOX11 SE regions following CRISPR-dCas9-KRAB targeting, compared to empty vector (sgEmpty) in KCNR cells. *P* values were generated from paired *t*-test across *n* = 3 replicates. Bars show the mean, error bars represent the standard deviation. **c** Bar graph of H3K27ac ChIP-qPCR showing a relative decrease in H3K27ac at specific SOX11 SE region following CRISPR-dCas9-KRAB targeting. sgRNAs targeting SOX4 SEs led to a significant decrease in H3K27ac peaks compared to empty vector (EV) in KCNR cells. *P* values were generated from paired *t*-test across *n* = 3 replicates.

Bars show the mean, error bars represent the standard deviation. **d** SOX11 mRNA levels (measured by RT-qPCR) decreased upon CRISPR-dCas9-KRAB mediated repression of SOX11 SEs in KCNR cells. Bars show the mean, error bars represent the standard deviation of *n* = 2 biological replicates. **e** Upper panel: Composite plots showing SOX11 (left) and H3K27ac (right) signal intensities (reads per million mapped reads) at SOX11 high confidence peaks in 4D EtOH (left) and 4D ATRA (right) treated cells. Middle and Lower panel: Heatmaps of SOX11 and H3K27ac peak intensity at SOX11 high confidence peaks. Each row represents a genomic location and is centered around SOX11 peaks, extended 2 kb in each direction, and sorted by SOX11 signal strength. **f** Line graph showing a decrease in expression of genes bound by SOX11 following 2, 4, and 8 days of ATRA treatment.

pioneer TFs that have important roles in developmental processes regulating cell fate and differentiation. In neural cells, the orchestrated yet transient expression and activity of SOX11 is crucial for the precise execution of a neurogenic program[49,50], it is needed to maintain progenitor pools and promote neuronal differentiation[51,52]. The sequential expression of different SOX genes has been shown to be important for early specification of the neural lineage in ES cells, with the subsequent expression of different SOX genes required for the transition to later stages of neural differentiation[49]. Little is known about the role of SOX genes in NB biology. In normal murine cells, Sox21 is required for Sox2-induced reprogramming of fibroblasts[53]. Our finding of the loss of the SOX21 SE and decreases in SOX21 mRNA expression suggest involvement in NB cell growth, although more detailed future studies are needed to examine the role of SOX21 in NB more fully. We focused our

studies on SOX11 and SOX4 since in developing sympathetic ganglia, Sox11 is essential for neuronal proliferation, but during development, SOX11 expression decreases as levels of SOX4 increase[40]. We found sequential changes in SEs driving SOX gene expression upon NB cell differentiation. We showed that SOX11 is expressed and contributes to their maintenance in a self-renewal or proliferating progenitor state, as evidenced by our results showing that loss of SOX11 leads to a marked reduction in both in vitro growth and NB tumor xenograft growth in mice (Fig. 7e). Upon differentiation, the SE driving SOX11 expression is lost while SE driving SOX4 levels is gained. Silencing of SOX11 is associated with an increase in the expression of differentiation genes, making NB permissive for differentiation but requiring additional stimuli such as the ATRA-induced gain of the SE driving SOX4 (Fig. 2), environmental factors such as NGF (Fig. S8g, lower panel) or other

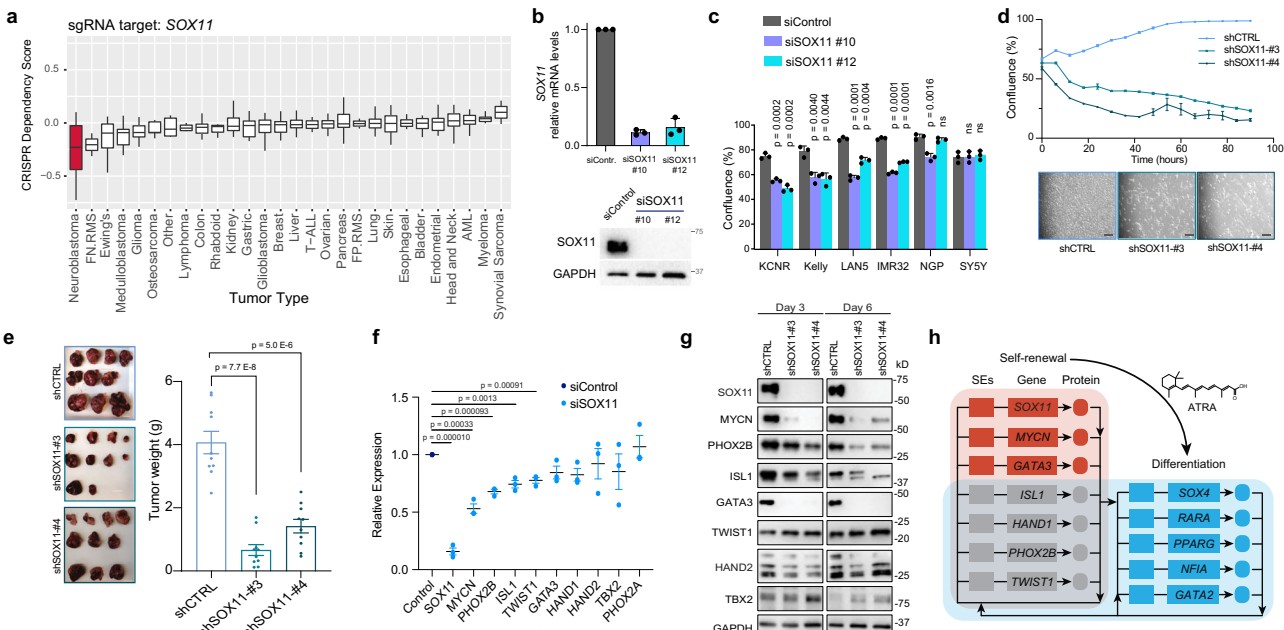

**Fig. 7 | SOX11 drives cell growth and proliferation of neuroblastoma cells. a** Bar graph showing the dependency of different cancer cells ·on SOX11 for cell growth and proliferation. NB cells are most preferentially dependent on SOX11 for growth and proliferation compared to other tumor types. Data mined from Project Achilles genome-wide CRISPR-Cas9 screen. The number of cell lines in each tumor type are listed in Supplementary Dataset 9. Box plots show a median with quartiles, and whiskers show the 1.5 × interquartile range. **b** Upper panel: qRT-PCR analysis showing relative mRNA levels of the SOX11 after silencing for 48 h. Bars show the mean ± SEM of three replicates. Lower panel: The western blot shows inhibition of SOX11 at the protein level. **c** Multiple NB cells demonstrate significantly decreased cell growth after siRNA-mediated silencing of SOX11. Graph showing representative data from $n = 3$ biological replicates. $P$ values were calculated with an unpaired, two-sided student's $t$-test across three technical replicates. Bars show the mean, with error bars representing the standard deviation. **d** Upper panel: Line graph showing decreased cell growth after shRNA-mediated genetic inhibition of SOX11 (shSOX11#3 and shSOX11#4) in KCNR cells. Data were presented as the mean with error bars showing the standard deviation. Lower panel: Representative images of KCNR cells following genetic silencing either with control (shCTRL) or

SOX11 specific shRNA (shSOX11#3 and shSOX11#4) at the 48 h time point. Scale bars are 200 μM. **e** Representative mouse tumor (left panel) and a bar graph (right panel) showing a significant decrease in tumor volume (weight) following shRNA-mediated genetic inhibition of SOX11. $P$ values were calculated with an unpaired, two-sided student's $t$-test with Welch's correction. Bars show the mean, with error bars representing the standard deviation. **f** RT-qPCR analysis showing relative mRNA levels of known NB regulatory TFs after silencing of SOX11 for 48 h. Graph showing representative data from three biological replicates. $P$ values were calculated with an unpaired, two-sided student's $t$-test with Welch's correction. The center line shows the mean, with error bars representing the standard deviation. **g** Western blot analysis showing protein levels of known NB regulatory TFs after 3 and 6 days of genetic inhibition of SOX11. **h** Model of CRC network swapping during ATRA-induced differentiation of NB. Genes highlighted were those which were consistently called in KCNR and LAN5 cells, both by enhancer analysis and gene expression changes. Lines represent inferred or verified binding by ChIP-seq, and do not necessarily indicate that such SE binding is an essential positive regulation event.

genes (Fig. 2). The finding in KCNR that overexpression of SOX4 alone was not sufficient to control NB growth or induce differentiation supports a requirement for multiple stimuli. The decrease of the SOX11 binding motif (and subsequent gain of the similar SOX4 binding motif) in SEs associated with NB cell differentiation (Fig. 3a–c) is consistent with a sequential or overlapping involvement of SOX genes model of differentiation. Our ChIP studies validated this, showing marked increases in SOX4 binding after ATRA treatment.

While this study has focused on the functional consequences of SOX11 and SOX4 during NB differentiation, we believe other genes, such as GATA3 and GATA2, also play key roles during this differentiation process. The SE driving GATA3 is also lost, while the SE driving GATA2 is gained during NB differentiation. GATA2 is a potent inhibitor of the proliferation of neuronal progenitors and indispensable for the differentiation of several tissues during embryogenesis[41]. Studies in other cancers show that depending on the type of cancer and cellular context, SOX4 functions as a tumor suppressor, exerting its effects through the induction of apoptosis[54,55]. Higher levels of GATA2 and SOX4 are found as NB cells differentiate, and in the tumors of NB patients' high levels of GATA2 or SOX4 are significantly associated with a better prognosis.

The cluster of lost SEs are linked to many genes involved in NB pathogenesis. LMO1 is known to synergize with MYCN to generate NB

in zebrafish[56] and also functions as a key coregulator of the neuroblastoma CRC[16]. The SE driving GATA3 expression is in cluster lost upon differentiation. The consequences of this GATA3 decrease impact LMO1 as the NB risk allele SNPrs2168101 G > T, which creates a novel GATA3 binding site leading to a SE driving the transcriptional regulator LMO1 (ref.38), is lost, leading to decreased LMO1 expression. The lost cluster also contains SEs linked to 8 TFs whose expression are significantly downregulated upon NB cell growth arrest and differentiation. The involvement of these TFs is supported by the findings that (1) high expression of six out of the eight genes (MYCN, ID2, TBX2, TWIST1, SOX21, and SOX11) is associated with poor prognoses in primary NB tumors (R2 database; SEQC dataset); (2) several are key components of or linked to the NB CRC (GATA3, TBX2, MYCN)[13,14,39], and (3) many functions as NB dependencies or are known to regulate NB growth and differentiation. TWIST1 is a direct target of MYCN[57] and regulates expression of the MYCN enhancer axis[58], while ID2, also a known inhibitor of neural differentiation in neuroblastoma and is regulated by MYCN[59,60]. Our results indicate that these genes contribute to the self-renewal capacity of NB cells as the loss of their SE reduces their expression and is critical for cells to fully implement a post-mitotic differentiation program.

ATRA treatment leads to a decrease in SOX11 expression by the loss of the SE driving SOX11. We find that SOX11 serves as an input to

the NB Core Regulatory Circuit (CRC) stimulating expression of MYCN and some CRC members but is not itself dependent on any other individual TF of the CRC but may be dependent on multiple CRC members. Our finding that SOX11 stimulates MYCN, but MYCN inhibits SOX11 expression suggests that SOX11 functions upstream of MYCN in a transcriptional regulatory path in which induction of a downstream target leads to feedback suppression of the upstream regulator. Thus, SOX11 functions via MYCN as an input into the adrenergic NB CRC (Fig. 7e), which is consistent with recent studies on SOX11 role in NB[61]. As the NB cells are rewired for differentiation by ATRA, we find five gained SEs driving differentiation-associated genes that are common between the two NB cell lines examined (Fig. 7e). While we have studied a functional role for SOX4 in this study, we do not believe SOX4 functions alone as overexpression of SOX4 is not sufficient to induce changes in cell growth nor increases in differentiation. The role of GATA2 and other genes will be explored in future studies. The recent finding that SOX11 may function as a pioneer TF[62] and its regulation of SWI/SNF components in NB[61] raises the possibility that SOX11 may have a unique function in NB cells developing along a sympathoadrenal differentiation pathway. Additionally, the contribution to differentiation of the 1st wave of gained SEs which is transiently changed, is unknown. GREAT analysis indicates these TFs are involved in responses to hormone stimuli, signaling events, and cell communication, but whether they are necessary, or bystander effects warrants further study.

In summary, we have identified distinct groups of cell-state-specific SEs driving processes consistent with changes needed to switch a self-renewal program to a differentiation program. In the future, it will be exciting to explore wider implications of SE circuitry creating potential energy barriers (Waddington mountains) locking cancer cells in de-differentiated states. The influence of small molecules on reshaping this landscape (as here) remains largely unexplored but will be a fruitful space to map the mechanisms by which cell identity is influenced by chemical and extracellular signals.

## Methods

### Cell lines

Human MYCN-amplified neuroblastoma (NB) cell lines [SMS-KCNR (KCNR), LAN5, IMR32, Kelly, and NGP] and MYCN-WT NB cell lines [SH-SY5Y (SY5Y)] were used in this study. SMS-KCNR is a cell line derived from bone marrow metastasis from a relapsed NB patient, which carries MYCN amplification; 1p36 LOH (loss of heterozygosity); ALKF1174L; ALT+; p53WT; H,K-,N-RAS WT. A number of our studies have detailed genetic and biologic changes associated with ATRA-induced differentiation of KCNR NB cells[25,63–67]. NB cell lines were obtained from the cell line bank of the Pediatric Oncology Branch of the National Cancer Institute and have been genetically verified. Cells were cultured in RPMI-1640 medium supplemented with 10% fetal bovine serum (FBS), 2 mM L-glutamine, 100 µg/mL streptomycin, and 100 U/ml penicillin at 37 °C in 5% $CO_2$. Human embryonic kidney cells (HEK293T) were obtained from ATCC and cultured in Dulbecco's modified Eagle's media (DMEM) with 10% FBS, 100 µg/mL streptomycin, and 100 U/ml penicillin. For control or SOX4 stable knockdown, SMARTvector Inducible Lentiviral shRNA (GE Healthcare) particles were generated using HEK293T cells. Stable knockdown KCNR cell line was generated by infecting KCNR cells with either non-targeting control shRNA lentiviral particles (SMARTvector Inducible Lentiviral shRNA GE Healthcare, Cat # VSC11707) or SOX4 shRNA lentiviral particles (Cat # V3SH11252-226882070, 227115611, 227170688) and selected using puromycin (500ug/ml, Sigma-Aldrich) to generate pools. Doxycycline (Dox, 0.125 ug/ml) was used to induce shRNA expression. For stable CRISPR-mediated super-enhancer knockdown KCNR cell lines, the sgRNAs (Supplementary Table 1) were cloned in pLVhU6-sgRNA hUbC-dCas9-KRAB-T2a-Puro (Addgene # 71236)[44] using Esp3I cut

site and sequenced for positive clones. Cells were transduced and selected as described above. Target inhibition was validated using RT-PCR 7 days after infection. All cell lines were routinely tested for and were free of mycoplasma. To generate stable Doxycycline-inducible SOX4 over-expressing KNCR cell lines, KCNR cells were transduced with concentrated lentivirus particles expressing pLVX-TetOne-SOX4-puro or pLVX-TetOne-CTRL-puro (empty vector). Then stable cell lines were selected and maintained in a complete RPMI medium supplemented with 0.5 mg/ml Puromycin.

### Differentiation assay

All-*trans* retinoic acid (ATRA) (Sigma-Aldrich (R2625)) dissolved in 95% ethanol was used as a differentiating agent. KNCR and LAN5 cells were plated at a density of $3.5 \times 10^6$ cells/150 mm dish and treated with ethanol or ATRA (5 µM). After 2, 4, and 8 of days treatment, the media was decanted, and cells were washed twice with ice-cold PBS and detached mechanically from the culture dish. Viable cell number was determined by trypan-blue exclusion and collected as separate aliquots for protein, RNA, and FACS analysis as previously described in ref. 64. For ChIP, cells were fixed at indicated time points, with 1% Formaldehyde solution (Thermo Scientific, Waltham, MA). For ATAC-seq, an aliquot of 50,000 cells under each condition was made. Cell confluency and neurite extension assays using Essen IncuCyte ZOOM or FLR were adopted to evaluate cell proliferation and differentiation in real-time.

### Transient transfection

For transient inhibition of transcription factors, NB cells were electroporated, using Nucleofector Solution V, Program A-30 for KCNR, LAN5, Kelly, NGP, and SY5Y cells and Nucleofector Solution L, Program C-005, in a Nucleofector II device (Amaxa Biosystems), with ON-TARGETplus siRNAs (Supplementary Table 2), according to the manufacturer's instructions. On-target plus non-targeting siRNA (5′-UGGUUUACAUGUCGACUAA-3′, Dharmacon) was used as a control. The SMARTpool and four independent SOX11 siRNAs were tested (Supplementary Table 3), out of which 90% inhibition was achieved for two siRNAs (#10 and #12) and were used for downstream studies.

Two different shRNAs targeting SOX11 (shSOX11-#3/pLKO.1, TRCN0000019177; shSOX11-#4/pLKO.1, TRCN0000019178) from Sigma-Aldrich and one non-targeting shRNA (shCTRL/pLKO.1, 5′-CCGGCCTAAGGTTAAGTCGCCCTCGCTCGAGCGAGGGCGACTTAACC TTAGGTTTTT-3′) were packaged into lentivirus. The supernatants containing lentivirus were harvested and concentrated using Lenti-X™ Concentrator (Takara, 631232). KCNR cells were transduced with concentrated lentivirus expressing shCTRL, shSOX11-#3, and shSOX11-#4. These KCNR shRNA cell lines were used for cell growth assays in vitro or animal experiments in vivo.

### In vivo animal experiment

NSG mice provided by NCI CCR Animal Resource Program were housed and treated under the protocol (PB-023) approved by the Institutional Animal Care and Use Committee of the National Cancer Institute. Briefly, 0.5 million KCNR cells, transduced with lentiviral shCTRL, shSOX11-#3 or shSOX11-#4 for 3 days, were injected into the fat pad of left adrenal gland of 5-week-old female NSG mice ($n = 10$ for each group). When the tumors in the shCTRL group reached 2 cm, all mice were sacrificed, and the tumors weighed.

### RNA isolation and RT-PCR

For RNA extraction, ~$2 \times 10^6$ cells were collected per biological replicate. Total RNA was isolated from NB cells after ethanol or ATRA treatment; siRNA inhibition, SOX4 knockdown, and SE knockdown, using RNeasy Plus Mini kit (Qiagen Inc.), as per the manufacturer's protocol and analyzed for integrity using an Agilent 2200 TapeStation

System (Agilent Technologies). Independent replicates (2–3) were performed for each group.

For quantitative PCR analyses, preparation of cDNA (ABI, Cat # 4387406) and real-time qPCR using Fast-SYBR green mix (ABI, Cat # 4385612) was done according to the manufacturer's protocol. Quantitative measurements of genes' levels were obtained using the BIO-RAD CFX Touch Real-time (RT) PCR detection system and performed in triplicate. Ct values were normalized to HPRT levels. Representative data from biological replicates were shown in this study. Gene expression as quantified by RT-PCR is expressed as relative expression compared to the basal expression under control or self-renewing state. The primers used for qPCR are listed in Supplementary Table 4. In doxycycline-induced shSOX4 cell lines, *SOX4* levels were inhibited by 50% after 4 days of DOX treatment. Following this, cells were treated for 4 days either with ethanol or ATRA, and gene expression was analyzed.

### RNA-sequencing

**Sample preparation and sequencing.** Total RNA was isolated and subjected to RNA-seq analysis from KCNR and LAN5 cells after 2, 4, and 8 days of ethanol or ATRA treatment, and KCNR cells transfected with siControl and siSOX11 (#10) for 48 h. Total RNA was extracted using the RNeasy Plus Mini kit (Qiagen Inc.), according to the manufacturer's protocol. Strand-specific whole transcriptome sequencing libraries were prepared using TruSeq® Stranded Total RNA LT Library Prep Kit (Illumina, San Diego, CA, USA), following the manufacturer's procedure. RNA-seq libraries were sequenced on a HiSeq2500 for paired-end with a read length of 126 bp or HiSeq 3000/4000 for paired-end with a read length of 150 bp.

**Data processing and analysis.** Briefly, the Fastq files were processed using Trimmomatic (version 0.30)[68] to remove low-quality bases. The trimmed fastq data were aligned to human genome hg19 with STAR (version 2.4.2a)[48] and annotated using UCSC or Ensemble file version 19 (Ensembl 74) using Partek® Flow® software, version 7.0 Copyright ©; 2020 Partek Inc., St. Louis, MO, USA. STAR software also generated the strand-specific gene read counts. About 77% of 70 million reads per sample were mapped to the human genome uniquely for a total of 90% mapping rate. To eliminate batch effect and identify differential gene expression, the gene reads count data from STAR were analyzed with R Package Deseq2[69]. Deseq2 uses methods to test for differential expression by use of a negative binomial generalized linear model and was also used to normalize the reads count data to generate z-scores for heatmap display. Heatmaps were created using heatmap.2 function in g plots (version 2.17.0).

**Pathway analysis.** Statistical results of differentially expressed genes from Deseq2 were analyzed using QIAGEN's Ingenuity® Pathway Analysis (IPA QIAGEN Inc.) Genes with false discovery rates (FDRs) <0.05 were used as input for IPA core analysis that identified significant canonical pathways.

**Gene set enrichment analysis.** Enrichment for curated gene sets (Molecular Signatures Database v 7.0, http://software.broadinstitute.org/gsea/msigdb/) was performed using GSEA software version 4.0[70,71]. Differential expression output from Deseq2 (comparing RNA-seq from perturbation with ethanol versus ATRA or control siRNA versus SOX11 siRNA) was used to rank genes according to the log$_2$ fold change. The GseaPreranked tool was employed with 1000 permutations, and only gene sets with a maximum list size of 500 were considered.

### Protein isolation and western blot analysis

Protein lysates from with or without ATRA-treated cells; siControl and siRNA inhibited cells, shControl and shSOX4 cells were extracted using RIPA buffer (50 mM Tris pH 8.0, 150 nM NaCl, 0.1% SDS, 0.5% sodium deoxycholate, and 1% Triton-X 100), containing 1X Protease and Phosphatase Inhibitor cocktail (Thermo Fisher, 78442). Protein concentration was measured using Bradford Assay (Biorad, Hercules, CA, 5000006). Protein (15–30 µg) was denatured in 4X Laemmli Sample Buffer (Biorad, Hercules, CA, #1610747), separated on 4–20% precast SDS gel (Biorad, Hercules, CA), and transferred on a nitrocellulose membrane. Membranes were incubated in a blocking buffer [5% dry milk in TBST (TBS with 0.2% Tween-20)] for 1 h, followed by overnight incubation with the primary antibody in a blocking buffer at 4 °C. Bands were visualized using enhanced chemiluminescence (Amersham Biosciences) or SuperSignal™ substrate (Thermo Fisher). The antibodies and the dilutions used in the study are listed in Supplementary Table 5.

### Cell cycle analysis by FACS

For determining changes in cell cycle progression before and after ATRA treatment, 2–5 million NB cells were harvested and fixed by resuspending them in an ice-cold solution containing 0.5 mL phosphate-buffered saline (PBS) and 4.5 mL of 70% EtOH. Following fixation, cells were pelleted by centrifuging for 5 min at 240 × g. Cells were resuspended in 1X PBS and pelleted again for 5 min at 240 × g. Residual PBS was decanted and cells were stained for 30 min at room temperature with propidium iodide (Miltenyi Biotech, San Diego, CA) in the presence of RNase. Stained cells were analyzed for DNA content by fluorescence-activated cell sorting using a FACScan flow cytometer (Becton Dickinson). Percentages of cells in the G1, S, or G2/M phases of the cell cycle were quantified with FlowJo software. For each sample, 20,000 events were collected. Three biological replicates were performed, and representative data were shown.

### ChIP-sequencing

**ChIP-seq reactions and antibodies.** KCNR and LAN5 cells with or without ATRA treatment were subjected to Chromatin immunoprecipitation (ChIP) using a ChIP-IT High Sensitivity kit (Active Motif) as per the manufacturer's protocol. Briefly, formaldehyde-fixed cells (-10 million cells) were flash-frozen and stored at −80 °C for later use. Nuclei were isolated, and the chromatin was sheared in the range of 200–1000 base pairs using either an Active Motif EpiShear Probe Sonicator (Active Motif). KCNR cells were sonicated at 30% amplitude pulse for 20 s on and 30 s off for a total sonication "on" time of 14 min, LAN5 cells were sonicated at 25% amplitude pulse for 20 s on and 30 s off for a total sonication "on" time of 13 min. Sheared chromatin samples were immunoprecipitated overnight at 4 °C with antibodies targeting H3K27ac (Active Motif, catalog #39133), H3K4me3 (Cell Signaling, catalog # 9751), H3K4me1 (Abcam, catalog # ab8895), and H3K9me3 (Active Motif, catalog # ab8898). DNA purifications were performed with the ChIP-IT High Sensitivity kit (Active Motif). To normalize the ChIP-seq signal, we used Active Motif ChIP-seq spike using *Drosophila* chromatin (Active Motif catalog # 53083) and an antibody against *Drosophila* specific histone variant H2Av (Active Motif, catalog #61686) according to the manufacturer's instructions. ChIP samples were subjected to qPCR using a ChIP-qPCR kit (Active Motif) as per the manufacturer's instruction. The primers are listed in Supplementary Table 6. ChIP-seq DNA libraries were prepared by Frederick National Laboratory for Cancer Research sequencing facility using Illumina TruSeq ChIP Library Prep Kit, after which DNA was size-selected with SPRIselect reagent kit (to obtain a 250–300 bp average insert fragment size). Libraries were multiplexed and sequenced using TruSeq ChIP Samples Prep Kit (75 cycles), cat. # IP-2-2-1012/1024 on an Illumina NextSeq500 machine. 35,000,000–45,000,000 reads were generated per sample.

**CUT&RUN reaction and antibodies.** CUT&RUN coupled with high-throughput DNA sequencing was performed using antibodies targeting SOX4 (Sigma, Catalog: HPA029901, Lot: D116454), SOX11 (Sigma, Catalog: HPA000536, Lot: BG117774), and Cutana pA/G-MNase

(Epicypher) according to the manufacturer's protocol. Briefly, SMS-KCNR cells ($5.0 \times 10^5$ per reaction) were washed and incubated with activated Concanavalin A beads for 10 min at room temperature. Cells were then resuspended in antibody buffer containing 0.01% digitonin, 1 mL of each antibody was added to individual cell aliquots, and tubes were rotated at 4 °C overnight. The following day, targeted chromatin digestion and release was performed with 2.5 mL Cutana pA/G-MNase and 100 mM $CaCl_2$. Retrieved genomic DNA was purified with the MinElute PCR purification kit and eluted in 10 mL of buffer EB. Sequencing libraries were prepared with the automated Swift 2S system, followed by 100bp-PE sequencing with Novaseq 6000.

### ChIP-seq data analysis

**Data processing.** ChIP-seq data analysis was done as previously described, using similar pipelines and parameters[72,73]. Briefly, ChIP-seq reads were aligned to reference the human genome (hg19) using BWA[74] to build version GRCh37 (hg19) of the human genome. SAM files were converted to BAM files using SAMTools[75] and sorted, de-duplicated with Picard, and indexed using SAMTools. To visualize ChIP-seq tracks in IGV, BAM files were converted to tdf files using IGV tools (version 2.3.57). ChIP-Seq peaks were detected using MACS2 (version 2.1.0)[76]. The default $p$ value cut-off for peak enrichment was set to $10^{-5}$ for all data sets. Peaks within 2500 bp to the nearest TSS were set as promoter-proximal, while all others were considered distal. The distribution of peaks (as intronic, intergenic, exonic, etc.) was annotated using HOMER. ChIP-seq read density values were normalized per million mapped reads for a particular region. Gene ontology was performed using GREAT[27], using hg19 and the whole genome as the background.

**Identification of enhancers and super-enhancers.** Enhancers were detected using H3K27ac as an enhancer mark. Active enhancers were ranked according to increasing H3K27ac levels. To detect super-enhancers, we used the modified ROSE2 algorithm (https://bitbucket.org/young_computation/rose)[77,78] to take care of MYCN amplification and ranked the enhancers that were identified using MACS. The peaks were ranked according to the H3K27ac peak intensity (length × density) with the input signal in the corresponding region subtracted. Peaks within 12.5 kb were stitched, however peaks within ±2000 bases of an annotated promoter were excluded from stitching. The point where $y = x$ was tangent to the curve formed by plotting the rank-ordered stitched enhancer was selected as the threshold separating super-enhancers (SE) from typical enhancers (TE). Each enhancer was associated with the RefSeq genes that either overlapped with the enhancer or whose transcriptional start sites (TSS) fell within 50 kb of the enhancer boundaries.

Differential peak calling between control and ATRA-treated samples was performed using BEDTools v2.25.0[79] in multicov mode to measure read counts, which were normalized per million mapped reads, and visualized using R package ggplot2 or NGS plot[80]. A Pearson correlation coefficient (r) was calculated to measure the linear correlation between the SE signal and mRNA expression of the downstream gene of interest.

**Identification of super-enhancer master group.** For downstream analysis, a master group of all called SEs from all time points and conditions was created by concatenating all SEs and merging those that overlapped. BEDTools intersect analysis was performed to identify the group of SEs that overlapped at least 50% and H3K27ac read count pileup was calculated in those SEs over the treatment time course.

**K-means clustering analysis.** Clustering of SEs in ATRA-treated KCNR cells was performed on the master group of all called SEs from all time points and conditions, and then calculating the H3K27ac read count pileup in each SE over the treatment time course. The variation and standard deviation in H3K27ac levels were then calculated across time

points and conditions. The SEs that remained stable over the course of treatment (SD <30%) were removed from the master group. Those whose temporal pattern changed under both control and ATRA conditions (Max variance <200 RPM H3K27ac) were also removed from downstream analysis. Then, each SE was normalized by Z-scoring the H3K27ac signal over all time points. K means clustering was performed, the elbow method to determine the optimal number of clusters, using the R statistical package base function kmeans (x, centers = 5). The clustering separated SEs into groups that lost signal or gained signal at different time points that we termed "waves" of SE activation.

**RPMPR data normalization.** For KCNR, we normalized H3K27ac according to reads per million peak reads (RPMPR) by first calling peaks with MACS2, then calculating the total number of reads only within peaks and normalizing to million reads within peaks.

**Linked enhancer-activated factor (LEAF) plots.** To create LEAF plots, we used the edge output from COLTRON[42] that reports for each $TF_a$ the list of other $TF_{(a...z)}$ where its motif is found in that TFs SE. Then, for each TF node, we extracted the expression level from RNA-seq data paired to the same cell line, treatment condition, and time point, and used expression to determine node rank and size. Then, arrows were drawn from edge connection data.

### Hi-C and data processing

KCNR cells for Hi-C were prepared following the user guide of the ARIMA GENOMICS Arima-Hi-C kit. Briefly, the cells were crosslinked and digested using an Arima restriction enzyme cocktail. The ends were filled and marked with biotin and ligated. After DNA shearing and size selection, biotin-labeled ligation junctions were enriched using streptavidin beads. Libraries were prepared using the Accel-NGS 2 S Plus DNA library kit (Swift Biosciences), following the manufacturer's protocol, which were then PCR-amplified. The resulting libraries were multiplexed and sequenced (150 bp pair-end) with the NovaSeq 6000 machine. Approximately 500 million reads were generated for each sample. The Hi-C-Pro (https://github.com/nservant/HiC-Pro) and Juicer pipeline (https://github.com/aidenlab/juicer) were used to process the Hi-C data, and a custom Arima-Hi-C-specific Juicer.sh command line script was integrated during this process. Juicebox was used to visualize and create the Hi-C maps.

### ATAC-seq

ATAC-seq was performed as previously described in refs. 81,82. Approximately 50,000 KCNR cells were pelleted, and the Tn5 transposition reaction was performed with the Nextera kit (Illumina) according to the manufacturer's protocol. ATAC libraries were sequenced on an Illumina NextSeq machine (paired-end 75-bp reads). The Fastq files were processed using Encode ATAC_DNase_pipelines (https://github.com/kundajelab/atac_dnase_pipelines) installed on the NIH biowulf cluster (https://hpc.nih.gov/apps/atac_dnase_pipelines.html).

### Reporting summary

Further information on research design is available in the Nature Portfolio Reporting Summary linked to this article.

## Data availability

The raw sequencing data and processed files were deposited in the GEO repository at the NCBI (GEO: GSE147408). Source data are provided with this paper.

## Code availability

The code used to analyze the data is all publicly available here: https://github.com/CBIIT/ChIP_seq and questions regarding its implementation can be directed to the corresponding authors.

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

## Acknowledgements

This work was funded by the Center for Cancer Research, Intramural Research Program at the National Cancer Institute, NIH. Berkley Gryder is funded by the DOD Peer Reviewed Cancer Research Program Career Development Award, the ALSF Crazy 8 Initiative, the ALSF "A" Award, the V Foundation, and the Angie Fowler AYA Cancer Research Initiative. We would like to thank Drs. Sridhar Hannenhalli, and Marielle Yohe for review of the manuscript. We are grateful to Drs. Jack Shern, Laura Kerosuo, and Mark Zimmerman for helpful discussions. We also thank Drs. Xinyu Wen and Haiyan Lei for data processing and Xiyuan Zhang for advice on bioinformatic analyses. We thank Bao Tran, Jyoti Shetty, and Yongmei Zhao from NCI Sequencing Facility for ChIP-DNA and RNA-sequencing.

## Author contributions

D.B., B.G. and C.J.T. designed the study, wrote the manuscript, and coordinated the entire study. D.B., S.B., Z.L., M.X. and M.S. performed the in vitro experiments and analyzed the data. D.B. performed the ChIP experiments, RNA-seq, and ATAC-seq experiments. Z.L. performed the Hi-C experiments. M.S. performed the in vivo experiment and analyzed the data. M.Z. performed the CUT&RUN of SOX4 and SOX11. B.G., H.-C.C. and D.B. performed bioinformatic analyses. B.G. and H.-C.C. generated custom codes and software for data analysis. S.A. performed statistical analysis. Z.V. and S.J.D. performed GWAS data

analysis. D.B., B.G., Z.L., S.A., S.J.D, J.K. and C.J.T. critically edited the manuscript. All the authors have reviewed the manuscript.

## Competing interests

The authors declare no competing interests.
