## [Peer Review File · Nature Communications]

REVIEWER COMMENTS

Reviewer #1 (Remarks to the Author):

To the Authors

In this manuscript, Banerjee et al, sought to identify the epigenetic changes that accompany neuroblastoma cell differentiation in response to retinoic acid through CHIP-seq analysis of the H3K27ac enhancer mark through various stages of ATRA treatment. Through integration with RNA-seq and Hi-C data, the authors report dynamic changes in the SE landscape that fluctuate in sequential waves following ATRA treatment, with 3 different stages or “clusters” of expressed genes. They further report that switching between lineage-specific SOX family TFs induces differentiation. The experiments use cutting-edge genomic techniques to infer enhancer landscape and enhancer-promoter interactions, followed by elegant validation studies using the CRISPR-KRAB repressor system. The concept is novel and interesting. The data largely support the conclusions and the narrative, except for the following that need to be further strengthened: 1) the analysis and biological significance of the fluctuating SE waves, 2) the evidence supporting the definition of the CRCs associated with the self-renewing and differentiated states and, most importantly, 3) the switch from SOX11 to SOX4 regulation in the differentiation state.

Major

1. One main issue is that the goal of the paper as stated in the introduction - to identify the epigenetic changes that accompany differentiation of NB cells during development - is not met through the experimental data shown. The results instead answer a different question – i.e. what are the enhancer-related alterations that occur when NB cells are exposed to ATRA. While this itself is important, as these presumably mirror what happens in patients treated with cis RA, as stated, it does not answer the authors’ goals. The authors thus need to revise their premise and their objective in the introduction.
2. Furthermore, the use of the term “self-renewing” is not clear. Have the authors (or other groups) shown that these NB cells have the capabilities of pluripotent cells, or is this term used loosely based on the general assumption that these cancer cells must be self-renewing? Please clarify.
3. The authors performed K-means clustering on SEs in ATRA-treated cells and list 4 SE clusters. While creating the matrix or master group of all SEs, how did they select the length of the SEs to compare the gained H3K27ac signal at all time points and conditions (\pm ATRA at 2, 4, and 8 days)? It would be better to present a heatmap of H3K27ac signals of all SEs for the four clusters. Moreover, the K-means clustering needs to be explained in detail in the Methods section.
4. After an initial decrease in the SE ranking at day 2, why does MYCN SE ranking go up at days 4 and 8 in the so-called second and third wave, such that in the third wave, MYCN again is the highest ranked SE (Fig. 1c)? On day 8, is MYCN re-expressed? Shouldn’t MYCN expression be suppressed in differentiated cells?

5. The SEs associated with the SOX4 TF are listed in both the first and second gained SE cluster in Fig. 2d. If this interpretation is correct, it would be important to state how these two different sets of SEs regulate the expression SOX4 at different time points.

6. A major issue is the inconsistency between the changes in mRNA and protein levels of the SE-regulated genes. The authors have extensively discussed the changes in enhancer regions and their chromatin states in ATRA-treated MYCN-amplified NB cells compared to control cells. However, none of the TFs that gain SEs and show increased RNA expression (ETS1, HIF1A, SMAD9, HEY1 (Fig. 2d) are upregulated at the protein level (Fig. S2f). In fact, ETS1 and HIF1A protein levels are actually decreased, and HEY1 and SMAD9 levels remain unchanged (Fig. S2f). Similar contradictions are also evident with genes in the 2nd wave.

These discrepancies raise a couple of important questions: (i) Is the proposed temporal regulation by SEs biologically relevant? Could it be possible that differentiation is driven mainly by loss of MYCN and SOX11, and the SE waves are bystander effects associated with ATRA-induced perturbation? (ii) Would the temporal profiles of the SEs and the associated downstream changes in gene/protein expression hold true if enhancers were mapped by deposition of other SE markers such as MED1, BRD4, or H3K4me1 (rather than just H3K27ac as in this study)? It is also possible that SEs alone do not regulate plasticity associated with cell state and these scenarios may be better explained by the promoter-centric interactions. For example, the H3K4me3 and ATAC-seq data could be used to explore the chromatin changes at the promoters and gene bodies of genes essential for differentiation. These data could be integrated with the RNA-seq data to identify sets of genes which are up- or down-regulated during the transition from the “self-renewal” to the differentiation state to identify differentiation signatures in NB cells. This could also potentially help to explain the discrepancies noted in the following point.

7. While the data for the inward binding at the SOX11 SE is very strong, the supposed increase in inward binding for the SOX4 SE is not convincing (35 in control vs 42 in ATRA-treated cells) (Fig. 3b). What are the 7 TFs that gain inward binding at the SOX4 SE after ATRA-treatment? Do any of them belong to the TFs that acquire SEs in the 1st and 2nd SE waves? Similarly, in the control cells also, SOX4 already has significant outward binding, which is consistent with its not insignificant expression in these cells.

8. Although it is clear that SOX4 protein levels increase on day 8 post-ATRA, there is also considerable expression of this protein in control cells as seen in Fig. S2f. How do changes in the associated SEs explain this observation? Please clarify.

9. In Fig. 4, although, the role of SEs in SOX4 regulation has been robustly validated using CRISPR perturbation and SOX4 overexpression in differentiation is shown, how SOX4 regulates the phenotypic and genotypic correlates of differentiation is not shown. In other words, (i) what are the transcriptional targets of SOX4 in control and ATRA-treated cells? SOX4 ChIP-seq +/- ATRA and correlation with gene expression changes (with or without SOX4 +/- ATRA) would answer this question. As SOX4 protein is already expressed at basal levels in control cells, it will be interesting to see whether its genomic distribution changes with increased protein levels in response to ATRA (ii) Why were DPYSL3 and TUBB3

chosen as validation targets? Are these direct targets of SOX4, i.e. do these have SOX4 binding sites in their regulatory regions?

10. The authors show that during ATRA-induced differentiation, SOX11 loses SEs while SOX4 gains SEs. However, it is not clear whether these changes in chromatin actually translate into 1) altered binding of these genes to regulatory elements of their target genes or 2) mRNA and protein expression in the target genes that are regulated by these two TFs. For example, are the SOX11-regulated genes decreased in expression in ATRA-treated cells (compared with control cells)? (Fig 6h- only looks at CRC genes, which is a rather selective approach). Conversely, do the SOX4-regulated genes change in expression in ATRA-treated cells? This is an important experiment as the control cells also have significant expression of SOX4. Thus, to claim that SOX4 is at the top of the differentiation hierarchy, it would be critical that genome-wide changes in chromatin binding of both SOX11 and SOX4 TFs in the individual states be performed, especially because the changes in SEs that bind to either gene are not significant. SOX11- and SOX4-based Hi-C on ATRA treated cells will further strengthen the data.

11. As the authors specifically focus on the LMO1 super enhancer, it would be useful to highlight the gene in the hockey plot in Fig. 1c for the readers to get an idea about its relative ranking as compared to other SEs. Despite substantial loss of GATA3 binding and loss of K27 acetylation, the changes in LMO1 mRNA levels are rather modest. Could it be due to the persistence of H3K4me3 marks at the promoter? How do LMO1 protein levels change with the ATRA time course?

12. The authors mention adrenergic and mesenchymal cell states without any comment/explanation as to why this is relevant to ATRA-induced differentiation. What happens to the adrenergic CRC upon ATRA treatment at both mRNA and protein levels? Does it lead to a complete collapse of the circuitry, or is part of the circuitry retained, as suggested by the stable TWIST1 protein levels in Fig.S2f?

13. With regards to defining the different CRCs, the authors need to first show the genome-wide H3K27ac ChIP-seq signals for the different clusters (metagene plots centered at the summits) for all typical enhancers/SEs in control and ATRA-treated cells. These genome-wide profiles should then be compared with those of CRC-associated TFs to determine how enhancer signals vary between the two datasets. A significant difference in CRC-associated TFs in different states would strengthen the data.

14. The authors need to describe how they arrived at the CRC for these cell states. Specifically, why is a part of the self-renewal CRC (ISL1, HAND1, PHOX2B and TWIST1) still retained during differentiation (Fig. 6h)? What is the experimental evidence that supports such a shared circuitry? Also, there is no significant change in expression of HIF1A or ETS1 protein after 4 days of ATRA, yet they are considered a part of CRC. Also, why is SOX4 part of the control or self-renewing cell CRC if the premise is that SOX4 SE is gained after ATRA treatment?

15. The location of SOX11 upstream of MYCN in the so-called self-renewing CRC is perhaps the most interesting and unexpected result of the entire story (Fig. 6g, h). Validation using shRNA or CRISPR-edited lines (SOX11 locus harboring genomic deletions) would strengthen this conclusion. Also, if this assertion is true, one would assume that SOX11 downregulation must precede MYCN downregulation upon ATRA

treatment. The authors should address this experimentally (using time- resolved RNA expression with spike-in controls performed at shorter time points i.e. Minor

1) In Fig. S1e, do the size and intensity of typical and super enhancers represent values from control or ATRA-treated cells? This should be mentioned clearly.

2) Were all the enhancer comparisons at different days made against the controls on day 2 only? This does not take into account whether there is any element of spontaneous differentiation (or not) of cells in culture for 4 or 8 days under normal conditions. Alternatively, the authors could have performed a pilot experiment to see if enhancers change in control ethanol-treated cells over 8 days of culture, and having found out that number of enhancers essentially remained constant, they chose day 2 for all the remaining experiments? If this is the case, please clarify in the text on p4 “The number of typical enhancers identified in the controls (EtOH-2D) essentially remained constant from 2 days....and decreased 20% by 8 days (n=7239)”.

3) For the ChIP-seq experiments in general, the authors need to provide details regarding the number of biological replicates per time point and correlation coefficients between replicates. For example, do the data showing fluctuations in the number of enhancers in Figs. S1g and h represent a typical experiment? Is the same pattern seen in other replicates?

4) Fig. 2: Please double check the % values of the SE-driven TFs in the four clusters in the text.

5) Fig. S2f: TBX3 is wrongly labelled as a 1st wave TF.

6) Also, the H3K4me1 and K4me3 marks were introduced without any explanation in the text, please clarify the goal of showing these tracks in the text.

7) The profile of ATRA-activated SOX4 SEs (intronic to CASC15) looks very different between Figs. 3d and 4d. If it is an issue with the axis range, please adjust accordingly so that there is parity between the figures.

8) Considering the data with dox-inducible SOX4 shRNA show only a modest effect on DPYSL3 and TUBB3 expression and neurite length (Fig. 4f-i), it would be important to know how the SOX4 CRISPR lines (sgSE.1 and 3) compare to empty vector-expressing lines in these assays. Also, in this figure and in other similar figures, please mention whether the results are SEM or SD of the number of replicates.

Reviewer #2 (Remarks to the Author):

The paper by Banerjee et al. describes the findings of evolving CRCs together with transcriptome and epigenomic changes upon ATRA induced differentiation of neuroblastoma cells. Four distinct patterns of super-enhancers were identified, one linked to stem cell development/specialization including MYCN, GATA3 and SOX11 and three other clusters related to neural development with functional evidence for a role for SOX4 in transition or maintenance of the differentiated state.

Comments:

This paper connects previous work on CRCs in adrenergic neuroblastomas towards how these CRCs are affected through ATRA induced differentiation. Gaining further insight into this process is of translational relevance in view of further improvement of differentiation therapies for high-risk neuroblastomas for which current therapeutic options are rather limited, in particular for relapsed cases. This paper describes 4 waves of dynamically altered SE-clusters upon ATRA treatment. The lost cluster typically includes the dependency factor (most notably MYCN and also GATA4 and ISL1). The other three clusters seem to undergo a temporal induction. Based on SE activity SOX11 and SOX4 are marked as most strongly differential active TFs in the lost and activated waves respectively. Further HiC analyses was performed in KCNR cells to investigate regulatory cis-DNA interactions at the presumed SE sites of the SOX4 and SOX11 genes. SOX11 dependency and role in control of differentiation was further investigated using in vitro knock down experiments. Taken together, this is a very interesting paper providing novel insights into the complex biology of a deadly pediatric cancer.

1. How do the proposed SE-clusters relate to the induced retino-sympathetic CRC reported recently by Zimmerman et al (which refers to the paper here under review), it appears that the proposed clusters of induced SEs are waves that are not sustained during longer ATRA treatment?

2. One of the SEs that is markedly increased upon ATRA treatment is linked to RET. As RET is critically important in neuronal differentiation (e.g. regulated by ALK, causing TRKA phosphorylation), it would be of interest to look also into RET more closely in relation to expression and possible functional connection with the ATRA induced phenotype.

3. KCNR was selected as main cell line for most of the experiments. It would be of interest for the less specialized reader to provide information on the genomic status of this cell line (MYCN/ALK/RAS-MAPK/hTERT/p53/... status) and refer to previous data on ATRA treatment and responsiveness for these cells to support the choice of this particular cell line.

4. The data for TBX2 upon ATRA treatment are intriguing and merit some more attention. The authors include TBX2 in the lost wave but when looking into the mRNA data TBX2 mRNA levels go up after initial ATRA treatment. Would it be possible to add western blot data for TBX2 in fig.S2?

5. Other apparent discrepancies between mRNA and protein levels that are not referred to are noted for TWIST1 and ETS1.

6. In fig.2F the H3K27ac peaks associated to MYCN appear to go up upon initial downregulation after ATRA treatment in KCNR in contrast to LAN5, nevertheless protein blots do show robust down regulation of MYCN at later time points despite H3K27ac being apparently restored. Perhaps another regulatory region is more functionally relevant for MYCN which is not captured by the K27ac analysis?

7. It would be interesting to dig a bit deeper into transcriptome data for the SOX4 rescue experiment (SOX4 KD upon ATRA treatment) for more in depth comparative analyses.

8. SOX11 dependency is explored using several publically available data sets and in vitro knock down analyses. Based on the current findings more in depth analyses of SOX11 overexpression or knock down in an in vivo context could provide additional insights into the critical role of SOX11 in neuroblastoma formation and maintenance.

9. The proposed insight for a distinct role for SOX11 in the adrenergic CRC is intriguing and could provide further support from SOX11 ChIP sequencing data.

10. Fig S1G: according to the manuscript and the GEO files available, H3K27ac ChIP-seq was performed at day 2,4 and 8, either with EtOH or ATRA. However, in the figure there is data shown for “day 0”. Why are these data not available in GEO? In case if these data are based on publically available data not generated for this manuscript, indicate this in the figure legend. In addition, it is not fully correct to compare the number of enhancers between day 0 and day 2, as there is no comparison for the expression of H3K27ac protein for day 0 and day 2, as shown in figure S1F for the other timepoints.

11. The strength of the data used in this manuscript is that both H3K27ac ChIP-seq as RNA-seq data are used upon ATRA treatment in KCNR cells. However, proximity analysis within topological domains (defined in which cells?) was used to associate SEs with their respective genes. According to material and methods, the RefSeq genes that either overlapped with the enhancer, or whose TSS fell within 50kb of the enhancer boundaries, were associated to the super-enhancer. Upon annotation, SE's are priorities based on associated with TF and correlation with RNA-expression data ($R > 0,45$, supplementary table 3), resulting in a limited number of SEs. Indeed, when having a look to the plots in figure 2C and S2E, the differences in expression for the annotated genes are low, even not visible, although significant. Possibly, many genes are wrongly annotated to a super-enhancer. Why are the genes in the first place not associated with the super-enhancer based on the RNA-seq and ATAC-seq data, instead of taking the gene with closest TSS? After this initial annotation, a further prioritization can be performed based on the correlation between the H3K27ac ChIP and RNA-seq data. The way the analysis is performed here probably results substantial misannotated SE-gene associations and thus loss of data.

12. Please clarify where the “lost super-enhancer” associated with SOX11 is located. There is discrepancy between Fig 3D and Fig 5A. The coordinates of the SOX11 downstream super-enhancer in Fig 3D are chr2:6427620-6453786. However, in Fig 5A and supplementary Fig 5A, the “lost super-enhancer” associated with SOX11 is marked more upstream. In Fig 5A, the super-enhancer defined in Fig 3D can be found where 2 orange stripes are next to each other. And according to the HiC data, the super-enhancer shown in Fig 3D is as such not interacting with the SOX11 locus. Why is the focus in Fig 3D not on the lost SE? In addition, while in Supplementary Fig 5A, a clear reduction in K27ac is observed, this is not the case for the ATAC-seq profiles. Please explain. Is there a possibility that the SE lost its activity (lost H3K27Ac) but chromatin is still open?

13. Fig 1C: Read depth can influence the position of the identified super-enhancers along the Y-axis, as higher read depth will result in increased H3K27ac signal. Please comment on read depth for every condition or possible normalisation strategies you used.

14. Fig 5D: The effect on SOX11 mRNA expression is limited. Possible several SEs need to be targeted rather than in order to achieve full reduction of SOX11 mRNA levels? This should be commented and if possible tested?

16. Supplementary Fig 5C: Based on only 2 non-MYCN amplified cell lines it is difficult to conclude the claim that MYCN predisposes the cells to SOX11 dependence. In contrast, this low dependency is probably more correlated to SOX11 expression. To address this, the dependency data could be correlated with SOX11 expression. In this context, it is important to note the CLBGA cell line has no MYCN overexpression but high SOX11 levels.

Minor:

- In general, more information is needed in the figure legends and at the figures: statistical test used (missing in Fig S2E, Fig 6D, Fig 6F-G), error bars and what they mean (missing in Fig S1B-C and S2E), number of replicates and if they are biological or technical (missing in Fig. S1A-C), timepoint of read-out (missing in Fig 6C and D, Fig S5D), normalisation strategy (Fig 2F), statistics (missing for Fig 4C LAN5 and Fig 4I)

- Legend Fig S1A: the word “decrease” is 2 times used in the sentence, please correct.

- Fig S1D: In the text, there is a reference to the genes MYCN, ISL1 and GATA3, but in the heatmap only MYCN and ASCL1 are depicted for the downregulated genes.

- Fig S2D: Indicate in figure legend that this is about TF-associated SEs. At this moment it is not clear what the difference is between Fig 1D and S2D.

- This sentence is difficult to read: “RNA-seq indicated changes in SEs signal of these TF associated SEs (Suppl. Fig. S2d) corroborated changes in the downstream gene expression (Suppl. Fig. S2e).”

- In the manuscript, there is no reference to Supplementary Figure 3.

- Please check annotation of the references to supplementary figures and be consistent: “Suppl” vs “Sup”.

- Some genes needed to be in italics are not.

- Figure legend Fig S5D concerns NTRK1, but NTRK1 is not marked in the figure. Actually, the same sentence is used in the legend as for Fig S5G...

- Concerning the sentence hereafter: “However, silencing of NB CRC components GATA3, PHOX2B, ISL2 and TBX2 (Suppl. Fig. 5g) did not lead to decreases in SOX11 (Fig. 6g)”. Why is there a reference to Suppl. Fig. 5g which shows NTRK1 expression upon SOX11 knockdown?

Reviewer #3 (Remarks to the Author):

This is a well written, interesting manuscript by Banerjee et al. focused on the transcriptional reprogramming that underlies the differentiation of neuroblastoma (NBL) cells with retinoic acid. Retinoic acid is a standard treatment for children with high-risk neuroblastoma, and there has been a renewed interest in differentiation therapy in cancer with the approval of drugs such as IDH inhibitors in leukemia. The work is timely and important.

The authors study the effects of ATRA in a time course in NBL cell lines by RNA sequencing and ChIP sequencing. They focus on the role of SOX11 as an oncogenic driver in NBL and the activation of SOX4 as important in the differentiation process.

All told, this is a strong manuscript although there are some questions to address before publication.

Figure 1.

The authors state that there is no change in the enhancer landscape in the EtOH controls over the time course compared to the ATRA treated cells, but they do not show the data. It would be important to show an overview of this data in the supplement, including the controls, to evaluate clustering of the samples (e.g., t-SNE plots in the space of TE or SE, which includes controls and ATRA treated samples).

There is a decrease in the SE at MYCN at 2 and 4 hours but it looks like it is regained in the 3rd wave. With chronic ATRA treatment, does MYCN increase at time points after 8 days? What happens to MYC in terms of expression or based upon SE/TE binding? Does MYC compensate for MYCN?

What are the motifs enriched in the subset of enhancers with a decrease in H3K27Ac binding over the time course?

Figure 2

How do the authors interpret the western blot in S2f for targets such as GATA2 and GATA3 where there is a pretty striking change in expression of these genes in the EtOH treated cells over time? For example,

while GATA3 is decreased compared to EtOH at each time point, GATA3 is more highly expressed even in the ATRA treated cells at 8D compared to control at 2D.

A table should be included showing the genes that are “validated” in the LAN5 cells that have similar changes in expression as in the KCNR cells with ATRA treatment in each cluster (Fig. 2e).

Since the focus of the paper is on SOX genes, the authors might comment in the discussion on their result that SOX21 also seems to be repressed in the lost cluster.

Figure 3

In Fig 3d, it is notable that the NBL models labeled adrenergic tend to have increased H3K27Ac load for SOX11 compared to the more mesenchymal cell lines, while the opposite is true for SOX4. The authors should comment on this. Are the adrenergic/mesenchymal signatures from previously published work (Von Groningen et al) enriched in the ATRA treated cells compared to the vehicle treated cells?

In the survival plots for SOX11, does SOX11 expression track with MYCN? Is this significant if corrected for MYCN or stage?

Figures 4 and 5 are strong with clear data and interpretations.

Is SOX4 Overexpression sufficient to induce differentiation? This experiment would further strengthen the work.

Figure 6.

The authors seek to place SOX4 and SOX11 as members of a CRC in NBL. In this case, they need more supportive data. They could use publically available ChIPseq data to demonstrate that other CRC members bind to enhancers for SOX11 and potentially SOX4 in NBL. This analysis should be done. Similarly, to definitively show Figure 6h, it would be important to demonstrate that SOX11 and SOX4 bind at enhancers for other CRC members. This reviewer recognizes the challenges of ChIP-seq when commercial antibodies are not effective. While potentially beyond the scope of the current manuscript, the authors could attempt ChIPseq with HA-tagged versions of the proteins, particularly for SOX11, or maybe rather tone down the claims that SOX11 sits atop the CRC hierarchy.

Are the SOX11 siRNA gene expression changes enriched for the ATRA signature by GSEA or some other test for statistical enrichment?

The authors might also comment that loss of HAND2 seems to increase SOX11 expression similar to the pattern seen for MYCN.

REVIEWER COMMENTS

Reviewer #1 (Remarks to the Author):

To the Authors

In this manuscript, Banerjee *et al*, sought to identify the epigenetic changes that accompany neuroblastoma cell differentiation in response to retinoic acid through ChIP-seq analysis of the H3K27ac enhancer mark through various stages of ATRA treatment. Through integration with RNA-seq and Hi-C data, the authors report dynamic changes in the SE landscape that fluctuate in sequential waves following ATRA treatment, with 3 different stages or “clusters” of expressed genes. They further report that switching between lineage-specific SOX family TFs induces differentiation. The experiments use cutting-edge genomic techniques to infer enhancer landscape and enhancer-promoter interactions, followed by elegant validation studies using the CRISPR-KRAB repressor system. The concept is novel and interesting. The data largely support the conclusions and the narrative, except for the following that need to be further strengthened: 1) the analysis and biological significance of the fluctuating SE waves, 2) the evidence supporting the definition of the CRCs associated with the self-renewing and differentiated states and, most importantly, 3) the switch from SOX11 to SOX4 regulation in the differentiation state.

Major

1. One main issue is that the goal of the paper as stated in the introduction - to identify the epigenetic changes that accompany differentiation of NB cells during development - is not met through the experimental data shown. The results instead answer a different question – i.e. what are the enhancer-related alterations that occur when NB cells are exposed to ATRA. While this itself is important, as these presumably mirror what happens in patients treated with cis RA, as stated, it does not answer the authors' goals. The authors thus need to revise their premise and their objective in the introduction.

We appreciate the reviewer pointing out that of the epigenetic changes we actually focused on were super enhancer related changes. Although we had indicated this in the Abstract as the focus of the studies, we agree with the reviewer and have now modified the Introduction to emphasize that we focused on the “super enhancer regulatory circuitry”.

2. Furthermore, the use of the term “self-renewing” is not clear. Have the authors (or other groups) shown that these NB cells have the capabilities of pluripotent cells, or is this term used loosely based on the general assumption that these cancer cells must be self-renewing? Please clarify.

We appreciate reviewer’s concern over the use of the term “self-renewing”. To clarify we have inserted “cancer cell” to now read “cancer cell self-renewal state” to specify it is in the context of an abnormal cell. We have included citations that present evidence that NB cells have capabilities and cell signatures of pluripotent cells and are self-renewing in nature. These studies show clear evidence that NB cells have capabilities of pluripotent cells and are self-renewing in nature.

3. The authors performed K-means clustering on SEs in ATRA-treated cells and list 4 SE clusters. While creating the matrix or master group of all SEs, how did they select the length of the SEs to compare the gained H3K27ac signal at all time points and conditions (\pm ATRA at 2, 4, and 8 days)? It would be better to present a heatmap of H3K27ac signals of all SEs for the four clusters. Moreover, the K-means clustering needs to be explained in detail in the Methods section.

We thank the reviewer for the suggestions that enhanced the clarity of the analysis process involved. The following changes were made in the manuscript. For the size of the super enhancers, we simply used the overlap across all enhancers identified in controls and 2,4, and 8 days after ATRA treatment. With the help of Bedtools, we identified the SEs that overlapped under all timepoints and conditions and then calculated the H3K27ac signal in each SE over the treatment time-course. We have now included in the Methods Section, sub-sections entitled “Identification of Super-Enhancer master group”.

As suggested by the reviewer, we have also included

- i. A heatmap of H3K27ac signals of all SEs for the four clusters in new Suppl. Fig. 2a.
- ii. A more detailed explanation for the K-means clustering has been added to the Methods section.

4. *After an initial decrease in the SE ranking at day 2, why does MYCN SE ranking go up at days 4 and 8 in the so-called second and third wave, such that in the third wave, MYCN again is the highest ranked SE (Fig. 1c)? On day 8, is MYCN re-expressed? Shouldn't MYCN expression be suppressed in differentiated cells?*

We thank the reviewer for raising this concern. By proximity analysis, MYCN is driven by 3 different SEs, indeed the SE signals are increased Day 8 compared to Day 2 and Day4, as shown in the table below. However, the H3K27ac levels of all 3 SEs at Day 8 still remains 1.4 to 5.4 fold below the levels in control cells. MYCN is not re-expressed at Day 8. As seen in Figure 2d, MYCN mRNA levels remain decreased at Day 8. In addition, MYCN protein levels stays low in differentiated cells at Day 8 as observed in Suppl. Fig. 3f. To make the visualization simpler we have highlighted the trend of the MYCN SE (SE3) which has the maximum signal. We have replaced the hockey plot with a newer version (Fig.1C). It is worth mentioning the height of the SE ranking above other TFs at 8 days is largely due to the fact that it is amplified, and the more appropriate comparison is the MYCN SE at day 0 compared to day 8.

					Fold Change in H3K27ac Signal		
	Chr	Start	End	Cluster	(ATRA/EtOH)		
					Day2	Day4	Day8
SE1	chr2	16054588	16055461	Cluster1	-2.46	-14.64	-5.49
SE2	chr2	16121837	16126743	Cluster1	-1.39	-2.75	-1.44
SE3	chr2	16330102	16426768	Cluster1	-1.27	-6.52	-2.22

5. *The SEs associated with the SOX4 TF are listed in both the first and second gained SE cluster in Fig. 2d. If this interpretation is correct, it would be important to state how these two different sets of SEs regulate the expression SOX4 at different time points.*

We appreciate the reviewer's comment. To address the issue, we have included a bubble plot depicting Pearson's correlation analysis (see table below) between H3K27ac levels of SOX4 SEs and RNA levels of SOX4 (Fig. 5, Panel J). Our analysis using Pearson's correlation shows that H3K27ac signals for both first and second wave SEs at Day 2 after ATRA treatment is positively correlated with SOX4 mRNA levels, whereas the RNA expression at Day 4 and Day 8 are guided by the second wave SE, and not the first

wave. We hypothesize that the first wave SE is driving the initial expression, and the second wave SE upregulated as a consequence of the long-range first wave SE.

	H3K27ac Signal		RNA-seq		Pearson
	EtOH	RA_Day2	EtOH	RA_Day2	
1st Wave	749	1898	66	165	1
2nd Wave	1337	1381	66	165	1
	H3K27ac Signal		RNA-seq		Pearson
	EtOH	RA_Day4	EtOH	RA_Day4	
1st Wave	749	735	66	159	-1
2nd Wave	1337	3871	66	159	1
	H3K27ac Signal		RNA-seq		Pearson
	EtOH	RA_Day4	EtOH	RA_Day8	
1st Wave	749	439	66	119	-1
2nd Wave	1337	3008	66	119	1

6. A major issue is the inconsistency between the changes in mRNA and protein levels of the SE-regulated genes. The authors have extensively discussed the changes in enhancer regions and their chromatin states in ATRA-treated MYCN-amplified NB cells compared to control cells. However, none of the TFs that gain SEs and show increased RNA expression (ETS1, HIF1A, SMAD9, HEY1 (Fig. 2d) are upregulated at the protein level (Fig. S2f). In fact, ETS1 and HIF1A protein levels are actually decreased, and HEY1 and SMAD9 levels remain unchanged (Fig. S2f). Similar contradictions are also evident with genes in the 2nd wave.

These discrepancies raise a couple of important questions:

We appreciate reviewer's concern over the discrepancies in RNA and protein levels of some TFs. We agree that there was not a direct correlation between all TFs mRNA and their level of protein expression. However, given that there are multiple levels of regulation at both the mRNA transcription and translation level as well as protein level, we don't believe it is unexpected. For this reason we decided to evaluate SOX11 and SOX4 as there was a correlation between mRNA and protein level. For example, SOX4 upregulated at both RNA and protein levels, also show increased binding on genes that are increased post ATRA treatment and involved in differentiation.

(i) Is the proposed temporal regulation by SEs biologically relevant? Could it be possible that differentiation is driven mainly by loss of MYCN and SOX11, and the SE waves are bystander effects associated with ATRA-induced perturbation?

Regarding the comment raised in (i), the reviewer raises an interesting point to which we believe this study is a first step towards addressing issues related to temporal regulation by SE of TFs involved in ATRA mediated NB cell differentiation. We would point out that the GREAT analysis is consistent with a sequential response as we might understand in the context of differentiation in response to a stimulus as we have noted in the 2nd line of the Discussion. The 1st Wave SE cluster involves transient responses to hormone stimuli, signaling events and cell communication and the role of their contribution to the differentiation process is not clear but is worthy of attention in future studies. We have included a comment in the discussion. We think not all of the temporal regulation by SEs are biologically relevant due to the contradictions between their RNA expression and protein levels. For this reason, in this study, we decided to evaluate SOX11 and SOX4 as there was a positive correlation between mRNA and protein level: SOX4 was upregulated at both RNA and protein levels, while SOX11 was decreased at both RNA and protein levels after ATRA treatment (Fig. 2f, Fig. S3f). Moreover, SOX4 and SOX11 are implicated in sympathoadrenal neural cell differentiation. In supporting that the temporal regulation of SOX4 and

SOX11 by SEs are biologically relevant, we found a consistent biological relevance (Fig. 3e) and biological effect (Fig. 4-7) of SOX4 and SOX11 in NB.

(ii) Would the temporal profiles of the SEs and the associated downstream changes in gene/protein expression hold true if enhancers were mapped by deposition of other SE markers such as MED1, BRD4, or H3K4me1 (rather than just H3K27ac as in this study)? It is also possible that SEs alone do not regulate plasticity associated with cell state and these scenarios may be better explained by the promoter-centric interactions. For example, the H3K4me3 and ATAC-seq data could be used to explore the chromatin changes at the promoters and gene bodies of genes essential for differentiation. These data could be integrated with the RNA-seq data to identify sets of genes which are up- or down-regulated during the transition from the “self-renewal” to the differentiation state to identify differentiation signatures in NB cells. This could also potentially help to explain the discrepancies noted in the following point.

Indeed, there are multiple ways to identify super-enhancers. In response to this question, we performed ChIP-seq in KCNR cells of H3K4me1 at 4 days +/- ATRA. We saw that changes were much less dramatic, as many SEs (as defined by H3K27ac data) were in fact premarked by H3K4me1. We exemplify this at the SE controlling SOX4 expression, central to the ATRA-induced phenomenon we are studying. In this study, we mainly focused on SEs but not promoters because SEs are known to play an important role in determining cancer cell identity. Consistently, here we find that the targeting of SOX4 SEs using CRSPRi results in a decrease of SOX4 expression and inhibition of RA mediated differentiation of KCNR cells (Fig. 4e). We agree that the promoter-centric interactions will be important in regulating plasticity associated with cell state, and we believe that the enhancer-promoter, SE-promoter interactions will play a key role. We plan to comprehensively address these questions in our future study.

Thus, in agreement with prior literature (2010, PNAS, Creighton et al, PMID: 21106759), we suggest that H3K27ac is better able to distinguish active enhancers, while H3K4me1 cannot distinguish between poised and active enhancer elements. For identification of genes which are up or downregulated during the transition, we chose to focus on SOX4/11 associated genes, instead of ATAC-seq or H3K4me3, and integrated RNA-seq.

7. While the data for the inward binding at the SOX11 SE is very strong, the supposed increase in inward binding for the SOX4 SE is not convincing (35 in control vs 42 in ATRA-treated cells) (Fig. 3b). What are the 7 TFs that gain inward binding at the SOX4 SE after ATRA-treatment? Do any of them belong to the TFs that acquire SEs in the 1st and 2nd SE waves? Similarly, in the control cells also, SOX4 already has significant outward binding, which is consistent with its not insignificant expression in these cells.

We recognize the reviewer's concern over the nominal increase in SOX4 binding sites. The TFs that gain inward binding at the SOX4 SE are increased in transcription over time. Three of these (NFIA, NFIB, PPARG) are also in either the first or second waves. The others change in gene expression but are not picked up as changing their SE's in a wave-like fashion, but nonetheless are likely influencing SOX4 via its SEs by increased binding. We have now included a bar graph of the 7 TFs that gain inward binding at the SOX4 SE in edited version Fig. 3, Panel c.

8. Although it is clear that SOX4 protein levels increase on day 8 post-ATRA, there is also considerable expression of this protein in control cells as seen in Fig. S2f. How do changes in the associated SEs explain this observation? Please clarify.-

We agree with reviewers comment that SOX4 has considerable expression at the protein levels in the control cells. However, our cut and run experiment analyzing SOX4 binding before and after ATRA treatment (new Fig. 5a-c), clearly shows increased SOX4 binding to DNA after ATRA treatment. This is included in the new data on SOX4 ChIP in the results section.

In Fig. 4, although, the role of SEs in SOX4 regulation has been robustly validated using CRISPR perturbation and SOX4 overexpression in differentiation is shown, how SOX4 regulates the phenotypic and genotypic correlates of differentiation is not shown. In other words, (i) what are the transcriptional targets of SOX4 in control and ATRA-treated cells? SOX4 ChIP-seq +/- ATRA and correlation with gene expression changes (with or without SOX4 +/- ATRA) would answer this question. As SOX4 protein is already expressed at basal levels in control cells, it will be interesting to see whether its genomic distribution changes with increased protein levels in response to ATRA (ii) Why were DPYSL3 and TUBB3 chosen as validation targets? Are these direct targets of SOX4, i.e. do these have SOX4 binding sites in their regulatory regions?

i. To address this important question, we performed SOX4 and SOX11 CUTandRUN (analogous to ChIP-seq) in KCN R cells treated with or without ATRA for 4 days. Our results showed little change overall in SOX11 binding, despite the loss of its super enhancer. For SOX4 however, we saw a clear increase in genome-wide binding. These new SOX4 sites were strongly enriched in the SOX4 motif, and were found near genes which on average increased in transcriptional output. These genes, by gene ontology enrichment analysis, were found enriched in GO biological terms including negative regulation of transferase activity, histone ubiquitination, glia cell migration, and noradrenergic neuron differentiation. We have now included this important finding in our manuscript in new Fig. 5, Panel a-d.

ii. DPYSL3 and TUBB3 are biochemical markers of neural differentiation as observed in previous studies from our lab (Choe et al, 2005, Veschi et al, 2017). And the ChIP Cut and Run experiment shows that indeed DPYSL3 and TUBB3 are direct targets of SOX4, with increased SOX4 binding after ATRA treatment. We have also included an IGV screenshot of SOX4 binding at these loci in new Fig. 5, Panel e.

10. The authors show that during ATRA-induced differentiation, SOX11 loses SEs while SOX4 gains SEs. However, it is not clear whether these changes in chromatin actually translate into 1) altered binding of these genes to regulatory elements of their target genes or 2) mRNA and protein expression in the target genes that are regulated by these two TFs. For example, are the SOX11-regulated genes decreased in expression in ATRA-treated cells (compared with control cells)? (Fig 6h- only looks at CRC genes, which is a rather selective approach). Conversely, do the SOX4-regulated genes change in expression in ATRA-treated cells? This is an important experiment as the control cells also have significant expression of SOX4. Thus, to claim that SOX4 is at the top of the differentiation hierarchy, it would be critical that genome-wide changes in chromatin binding of both SOX11 and SOX4 TFs in the individual states be performed, especially because the changes in SEs that bind to either gene are not significant.- SOX11- and SOX4-based Hi-C on ATRA treated cells will further strengthen the data.

We appreciate reviewer's concern over the role of SOX11 and SOX4 in ATRA mediated differentiation. As mentioned in the previous comment to address this important question, we performed SOX4 and SOX11 CUTandRUN (analogous to ChIP-seq) in KCNR cells treated with or without ATRA for 4 days. Our results showed little change overall in SOX11 binding, despite it losing its super enhancer. Furthermore, at many SOX11 bound genes that showed no decrease in SOX11 binding, we still observe downregulation (n = 182 genes) mediated by ATRA (Fig. 6e,f).

For SOX4 however, we saw a clear increase in genome-wide binding. These new SOX4 sites were strongly enriched in the SOX4 motif, and were found near genes which on average increased in transcriptional output. These genes, by gene ontology enrichment analysis, were found enriched in GO biological terms including negative regulation of transferase activity, histone ubiquitination, glia cell migration, and noradrenergic neuron differentiation. We have now included this important finding in our manuscript in new Fig. 5, Panel a, b and c.

We agree that the SOX11- and SOX4-based Hi-C on ATRA treated cells will further strengthen the data, however, we don't have good ChIP-seq grade SOX11- and SOX4- antibodies and such antibodies are needed for Hi-C. This is also the reason why we performed CUT&RUN instead of traditional ChIP-seq to identify SOX4 and SOX11 binding sites.

11. As the authors specifically focus on the LMO1 super enhancer, it would be useful to highlight the gene in the hockey plot in Fig. 1c for the readers to get an idea about its relative ranking as compared to other SEs. Despite substantial loss of GATA3 binding and loss of K27 acetylation, the changes in LMO1 mRNA levels are rather modest. Could it be due to the persistence of H3K4me3 marks at the promoter?

How do LMO1 protein levels change with the ATRA time course?

We thank the reviewer for the suggestion. We have edited the hockey plot to mark the relative position of LMO1 SE. When we do see a loss in GATA3 binding and H3K27ac levels after ATRA treatment, however, a closer look at the H3K27ac marks shows that there are still residual H3K27ac signals at LMO1 SE and promoter regions. We hypothesize, this is due to other TFs that are not significantly affected by ATRA treatment, lead to LMO1 expression, albeit reduced. Western blot analysis shows there is a slight change in the LMO1 protein levels with ATRA at 2 and 4 days.

12. The authors mention adrenergic and mesenchymal cell states without any comment/explanation as to why this is relevant to ATRA-induced differentiation. What happens to the adrenergic CRC upon ATRA

treatment at both mRNA and protein levels? Does it lead to a complete collapse of the circuitry, or is part of the circuitry retained, as suggested by the stable TWIST1 protein levels in Fig.S2f?

We appreciate the reviewer's comment. Our clustering analysis of NB cells utilizing published H3K27ac ChIP-seq data from Von Groningen et al, 2017, Nat Genet, and H3K27ac data from KCNR cells, shows KCNR cell line are adrenergic in nature and retains its characteristic even after ATRA treatment, as seen in the correlation plot and ADRN signature heatmap below. Both control cells and ATRA treated cells cluster along with other ADRN cell lines based on published literature.

Looking into the changes in protein expressions of some key genes of adrenergic CRC (PHOX2B, PHOX2A, SOX11, GATA3, GATA2, HAND1, ASCL1) we observe decreases in SOX11, GATA3, PHOX2A, PHOX2B, ASCL1 and increases in GATA2, HAND1 expression

We have now included the clustering analysis and heatmap showing enrichment of ADRN signature in our revised Suppl. Fig. 5.

13. With regards to defining the different CRCs, the authors need to first show the genome-wide H3K27ac ChIP-seq signals for the different clusters (metagene plots centered at the summits) for all typical enhancers/SEs in control and ATRA-treated cells. These genome-wide profiles should then be compared with those of CRC-associated TFs to determine how enhancer signals vary between the two datasets. A significant difference in CRC-associated TFs in different states would strengthen the data. –

To perform this, we first took the clustered super enhancers (which were regions called as “super” in any of the conditions or timepoints surveyed) and identified the summits of individual H3K27ac peaks within them. Then, as suggested here, we plotted the metagene plots genome-wide for H3K27ac ChIP-seq signal for the different clusters. Typical enhancers are mixed in with SEs, because many of the enhancers in question are dynamically changing enhancer category. Then, we compared these with those of CRC-associated TFs. The results, shown below, illustrate the genome-wide flux of H3K27ac signal both at all SEs and at CRC-associated H3K27ac sites.

14. The authors need to describe how they arrived at the CRC for these cell states. Specifically, why is a part of the self-renewal CRC (ISL1, HAND1, PHOX2B and TWIST1) still retained during differentiation (Fig. 6h)? What is the experimental evidence that supports such a shared circuitry? Also, there is no significant change in expression of HIF1A or ETS1 protein after 4 days of ATRA, yet they are considered a part of CRC. Also, why is SOX4 part of the control or self-renewing cell CRC if the premise is that SOX4 SE is gained after ATRA treatment?

It is important that some CRC members are involved in both self-renewal and differentiated states, otherwise the energy barrier would be too high for the transition. The experimental evidence for their inclusion is that the SEs controlling these genes, and the RNA-seq for them, are intact in differentiated cells. Regarding SOX4, please take a closer look at the CRC diagram in the final figure panel (new Fig. 7h)- SOX4 was not listed as part of the self-renewing CRC, but rather as part of the differentiation CRC.

15. The location of SOX11 upstream of MYCN in the so-called self-renewing CRC is perhaps the most interesting and unexpected result of the entire story (Fig. 6g, h). Validation using shRNA or CRISPR-edited lines (SOX11 locus harboring genomic deletions) would strengthen this conclusion. Also, if this assertion is true, one would assume that SOX11 downregulation must precede MYCN downregulation upon ATRA treatment. The authors should address this experimentally (using time-resolved RNA expression with spike-in controls performed at shorter time points i.e. <day 2) to bolster the claim for positioning SOX11 on top of CRC hierarchy.

As suggested by the reviewer, we validated regulation of MYCN by SOX11 using SOX11 shRNA cell lines. We found inhibition of SOX11 led to downregulation of MYCN protein levels (Figure 7, panel g) along with other members of the CRC (PHOX2B, HAND2, ISL1, TWIST1). This strengthened our model that SOX11 is upstream of MYCN in the self-renewing cells. However, using time-resolved RNA expression with ATRA done at shorter time points (12hrs, 24hrs, 48hrs), we were not able to conclude whether SOX11 downregulation preceded MYCN downregulation.

Minor

1) In Fig. S1e, do the size and intensity of typical and super enhancers represent values from control or ATRA-treated cells? This should be mentioned clearly.

As suggested by the reviewer, in Results Section, we have mentioned that the size and intensity of typical and super enhancers represent values from control KCNR cells.

2) Were all the enhancer comparisons at different days made against the controls on day 2 only? This does not take into account whether there is any element of spontaneous differentiation (or not) of cells in culture for 4 or 8 days under normal conditions. Alternatively, the authors could have performed a pilot experiment to see if enhancers change in control ethanol-treated cells over 8 days of culture, and having found out that number of enhancers essentially remained constant, they chose day 2 for all the remaining experiments? If this is the case, please clarify in the text on p4 "The number of typical enhancers identified in the controls (EtOH-2D) essentially remained constant from 2 days....and decreased 20% by 8 days (n=7239)".

We appreciate the reviewer's concern regarding the use of control on day 2 for all comparison. As suggested by the reviewer, we had performed a pilot experiment to see if enhancers change in control ethanol-treated cells over 8 days of culture and found out that number of super-enhancers do not change significantly from day 2 to day 8 in controls. Additionally, as described in the Materials and Methods, Section: K-means clustering analysis Page 30, of the revised manuscript, we also utilized the control data from day 2 to day 8 for calculating the H3K27ac read count pileup in each SE over the treatment time-course. The variation and standard deviation in H3K27ac levels were calculated across timepoints and conditions. The SEs that remained stable in controls and ATRA treated cells over the course of treatment (SD < 30%) were removed. Further, those whose temporal pattern changed under both control and ATRA conditions (Max variance <200 H3K27Ac) were also removed from downstream analysis. We have now reordered the sentence "The SEs in controls essentially remained constant over time (data not shown)." in the Result, Sections - All-trans retinoic acid (ATRA) mediated differentiation leads to dynamic changes

in super-enhancer landscape of MYCN-amplified NB cells, Page 4 in our revised manuscript for better clarification.

3) For the ChIP-seq experiments in general, the authors need to provide details regarding the number of biological replicates per time point and correlation coefficients between replicates. For example, do the data showing fluctuations in the number of enhancers in Figs. S1g and h represent a typical experiment? Is the same pattern seen in other replicates? –

We appreciate reviewer's comment. For our ChIP-seq experiments in this paper instead of doing biological replicates we focused on validating our observations with TFs in additional NB cell line, i.e., LAN5. As we focused primarily on identifying TFs crucial for NB growth and differentiation, in the current study though we are limited on our knowledge regarding the pattern in biological replicates, our study does confirm the dynamic expression of the TFs across 2 NB cell lines.

4) Fig. 2: Please double check the % values of the SE-driven TFs in the four clusters in the text. We thank the reviewer for pointing this out. We have now edited the values in the text and have also provided the actual numbers.

5) Fig. S2f: TBX3 is wrongly labelled as a 1st wave TF.

We thank the reviewer for pointing out the inadvertent mistake. We have now corrected the error.

6) Also, the H3K4me1 and K4me3 marks were introduced without any explanation in the text, please clarify the goal of showing these tracks in the text.

We appreciate reviewer's comment and have now clarified the goals of these tracks in the text of our revised manuscript.

7) The profile of ATRA-activated SOX4 SEs (intronic to CASC15) looks very different between Figs. 3d and 4d. If it is an issue with the axis range, please adjust accordingly so that there is parity between the figures.

We have zoomed in on the SE in Figure 4 to only show the portion of SOX4 distal regulatory region that was shown in Figure 3. These now have better visual parity.

8) Considering the data with dox-inducible SOX4 shRNA show only a modest effect on DPYSL3 and TUBB3 expression and neurite length (Fig. 4f-i), it would be important to know how the SOX4 CRISPR

lines (sgSE.1 and 3) compare to empty vector-expressing lines in these assays. Also, in this figure and in other similar figures, please mention whether the results are SEM or SD of the number of replicates.

We appreciate reviewer's suggestion and evaluated the effect of SOX4 inhibition on DPYSL3 and TUBB3 in the SOX4 CRISPR lines. In both sgSE.1 and sgSE.3 lines, DPYSL3 and TUBB3 mRNA levels after ATRA treatment are significantly reduced with sgRNA mediated decrease in SOX4 mRNA levels, as observed in dox-inducible SOX4 shRNA lines. We have included the data in revised Fig. 4, panel e (middle and right panel) and have also edited the text in our revised manuscript (Section: Gained cluster super-enhancer contributes to increased SOX4 mRNA).

Reviewer #2 (Remarks to the Author):

The paper by Banerjee et al. describes the findings of evolving CRCs together with transcriptome and epigenomic changes upon ATRA induced differentiation of neuroblastoma cells. Four distinct patterns of super-enhancers were identified, one linked to stem cell development/specialization including MYCN, GATA3 and SOX11 and three other clusters related to neural development with functional evidence for a role for SOX4 in transition or maintenance of the differentiated state.

Comments:

This paper connects previous work on CRCs in adrenergic neuroblastomas towards how these CRCs are affected through ATRA induced differentiation. Gaining further insight into this process is of translational relevance in view of further improvement of differentiation therapies for high-risk neuroblastomas for which current therapeutic options are rather limited, in particular for relapsed cases. This paper describes 4 waves of dynamically altered SE-clusters upon ATRA treatment. The lost cluster typically includes the dependency factor (most notably MYCN and also GATA4 and ISL1). The other three clusters seem to undergo a temporal induction. Based on SE activity SOX11 and SOX4 are marked as most strongly differential active TFs in the lost and activated waves respectively. Further HiC analyses was performed in KCNR cells to investigate regulatory cis-DNA interactions at the presumed SE sites of the SOX4 and SOX11 genes. SOX11 dependency and role in control of differentiation was further investigated using in vitro knock down experiments. Taken together, this is a very interesting paper providing novel insights into the complex biology of a deadly pediatric cancer.

1. How do the proposed SE-clusters relate to the induced retino-sympathetic CRC reported recently by Zimmerman et al (which refers to the paper here under review), it appears that the proposed clusters of induced SEs are waves that are not sustained during longer ATRA treatment?

We thank the reviewer for asking this thoughtful question. To answer this question, we looked in the expression of the important CRC factors reported by Zimmerman et al and presented them in the bar graph below.

We compared the expression of the genes 4 days after ATRA treatment from the current study to the expression of these genes after 6 days of ATRA treatment from Zimmerman et al study. Both studies showed upregulation in RARA, RARB, SOX4, TBX3, TBX2, and HIC1. No significant difference in expression was seen in HAND2 and ISL1 expression. Among the genes of the induced retino-sympathetic CRC, our study did not show RARB, MEIS1, and HIC1 being driven by SEs. We observed TBX2 was part of the Lost cluster, whose initial upregulation was followed by reduced expression at Day 8. The TFs SOX4, RARA, TBX3 were part of gained cluster. However, our study did not show RARB, MEIS1, and HIC1 being driven by SEs. So taken together data show strong overlap between the studies despite use of different cell lines and varied treatment times.

2. One of the SEs that is markedly increased upon ATRA treatment is linked to RET. As RET is critically important in neuronal differentiation (e.g. regulated by ALK, causing TRKA phosphorylation), it would be of interest to look also into RET more closely in relation to expression and possible functional connection with the ATRA induced phenotype.

We agree with the reviewer that given RET important role in neuronal differentiation and its dysregulation in neuroendocrine tumors indicates that it is an important gene to evaluate. However, we believe it is beyond the scope of the manuscript at this time (TFs focused) and anticipate that researchers' with an interest in RET will investigate in the future.

3. *KCNR was selected as main cell line for most of the experiments. It would be of interest for the less specialized reader to provide information on the genomic status of this cell line (MYCN/ALK/RAS-MAPK/hTERT/p53/... status) and refer to previous data on ATRA treatment and responsiveness for these cells to support the choice of this particular cell line.*

We have been studying issues related to differentiation with SMS-KCNR for more than 30years in our laboratory. We have included the requested information (below) on SMS-KCNR in the Materials and Methods (p25, lines 642-645).

SMS-KCNR is a cell line derived from bone marrow metastasis from a relapsed NB patient, which carries MYCN amplification; 1p36 LOH(loss of heterozygosity); ALK1174L;ALT+; p53WT; H,K-,N-RAS WT. A number of our studies have detailed genetic and biologic changes associated with ATRA induced differentiation of KCNR NB cells (PMID:3855502; PMID:3060792; PMID:1322787; PMID:8394722; PMID:9334253; PMID:12700651).

4. The data for TBX2 upon ATRA treatment are intriguing and merit some more attention. The authors include TBX2 in the lost wave but when looking into the mRNA data TBX2 mRNA levels go up after initial ATRA treatment. Would it be possible to add western blot data for TBX2 in fig.S2?

Thank you for the suggestion. We have included a western blot in our new Suppl. Fig S3f showing changes in TBX2 protein levels after ATRA treatment. We agree with the reviewer that we do see some increases in TBX2 expression after 2-4 days ATRA treatment, despite being a part of the lost cluster. Given that the mRNA levels of TBX2 after 8D ATRA is lower than the control cells, the clustering analysis calls it a candidate of lost cluster. We agree that it is an interesting gene. However, given the number of interesting genes identified in this screen, we focused on 2, SOX11 and SOX4. We have an interest in TBX for our future studies and feel the publication of this study will enable other researchers with an interest in this gene to utilize this information as well.

5. Other apparent discrepancies between mRNA and protein levels that are not referred to are noted for TWIST1 and ETS1.

We appreciate reviewer's concern over the discrepancies in RNA and protein levels of TWIST1 and ETS1. It is possible that there are multiple levels of regulation of these TFs at both the mRNA transcription and translation level as well as protein level to compensate the gain or loss of these genes after ATRA treatment. It is possible that some of the temporal regulation by SEs is not biologically relevant due to the contradictions between their RNA expression and protein levels. For these reasons as described in the manuscript, here we decided to evaluate SOX11 and SOX4 as there was a positive correlation between mRNA and protein level: SOX4 was upregulated at both RNA and protein levels, while SOX11 was decreased at both RNA and protein levels after ATRA treatment (Fig. 2f, Fig. S3f). We have added this in discussion.

6. In fig.2F the H3K27ac peaks associated to MYCN appear to go up upon initial downregulation after ATRA treatment in KCNR in contrast to LAN5, nevertheless protein blots do show robust down regulation of MYCN at later time points despite H3K27ac being apparently restored. Perhaps another regulatory region is more functionally relevant for MYCN which is not captured by the K27ac analysis?

					Fold Change of H3K27ac Signal		
	Chr	Start	End	Cluster	(ATRA/EtOH)		
					Day2	Day4	Day8
SE1	chr2	16054588	16055461	Cluster1	-2.46	-14.64	-5.49
SE2	chr2	16121837	16126743	Cluster1	-1.39	-2.75	-1.44
SE3	chr2	16330102	16426768	Cluster1	-1.27	-6.52	-2.22

We thank the reviewer for raising this concern. By proximity analysis, MYCN is driven by 3 different SEs, indeed the SE signals are increased Day 8 compared to Day 2 and Day4, as shown in the table above. However, the H3K27ac levels of all 3 SEs at Day 8 still remains 1.4 to 5.4 fold below the levels in control

cells. As noted by the reviewer, MYCN is not re-expressed at Day 8. As seen in Figure 2d, MYCN mRNA levels remain decreased at Day 8. In addition, MYCN protein levels stays low in differentiated cells at Day 8 as observed in Suppl. Fig. S3f. Previous studies have shown that MYCN also undergoes post-translational modifications, that lead to its degradation. A recent review on MYCN from the laboratory by Liu *et al*, describes it as “A major signaling pathway affecting MYCN protein stability occurs upon activation of PI3K. PI3K activates Akt which phosphorylates GSK3 β , suppressing GSK3 β kinase activity. This results in decreased phosphorylation of MYCN-T58 which is critical for targeted degradation by the proteasome”.

However, to make the visualization simpler and to incorporate additional changes suggested by reviewers, we have replaced the hockey plot with a newer version.

7. It would be interesting to dig a bit deeper into transcriptome data for the SOX4 rescue experiment (SOX4 KD upon ATRA treatment) for more in depth comparative analyses. -

To address this, we performed RNA-seq in triplicate for these same KCNR cells with Dox inducible shRNA against SOX4, treated with EtOH, or 1.7 μ M or 5 μ M of ATRA. The results, presented here, show that beyond just the 2 genes we previously performed RT-qPCR for (TUBB3 and DPYSL3), the larger Mesenchymal signature pathway is both activated by RA and rescued by SOX4 knockdown. This is now presented in new Figure 5j:

8. SOX11 dependency is explored using several publicly available data sets and in vitro knock down analyses. Based on the current findings more in depth analyses of SOX11 overexpression or knock down in an *in vivo* context could provide additional insights into the critical role of SOX11 in neuroblastoma formation and maintenance.

We thank the reviewer for the valuable suggestion. As per the suggestion we generated two SOX11 shRNA cell lines. Both shRNAs led to 90% reduction in SOX11 protein levels with approximately 80% decrease in cell confluency 6 days post-transduction. As detailed in the revised manuscript we have shown that loss of SOX11 led to a decrease in tumor volume. We have now included this data in Figure 7, Panel e.

9. The proposed insight for a distinct role for SOX11 in the adrenergic CRC is intriguing and could provide further support from SOX11 ChIP sequencing data.

We appreciate reviewer's thoughts on the role of SOX11 in adrenergic CRC. As mentioned in the previous comments to address this important question, we performed SOX11 CUTandRUN (analogous to ChIP-seq) in KCNR cells treated with or without ATRA for 4 days. Our results showed little change overall in SOX11 binding, despite it losing its super enhancer. However, at many SOX11 bound genes even though there was no decrease in SOX11 binding, we observed decreased expression (n = 182 genes) after ATRA treatment. We have now included SOX11 Cut and data in revised Fig 6, Panels e,f.

10. Fig S1G: according to the manuscript and the GEO files available, H3K27ac ChIP-seq was performed at day 2,4 and 8, either with EtOH or ATRA. However, in the figure there is data shown for “day 0”. Why are these data not available in GEO? In case if these data are based on publically available data not generated for this manuscript, indicate this in the figure legend. In addition, it is not fully correct to compare the number of enhancers between day 0 and day 2, as there is no comparison for the expression of H3K27ac protein for day 0 and day 2, as shown in figure S1F for the other timepoints.

We are unsure why there was any confusion on this point - we in fact did not perform any day 0 experiments other than neurite length (Fig S1C); EtOH control experiments for S1G were at day 2 only.

11. The strength of the data used in this manuscript is that both H3K27ac ChIP-seq as RNA-seq data are used upon ATRA treatment in KCNR cells. However, proximity analysis within topological domains (defined in which cells?) was used to associate SEs with their respective genes. According to material and methods, the RefSeq genes that either overlapped with the enhancer, or whose TSS fell within 50kb of the enhancer boundaries, were associated to the super-enhancer. Upon annotation, SE's are priorities based on associated with TF and correlation with RNA-expression data ($R > 0.45$, supplementary table 3), resulting in a limited number of SEs. Indeed, when having a look to the plots in figure 2C and S2E, the differences in expression for the annotated genes are low, even not visible, although significant. Possibly, many genes are wrongly annotated to a super-enhancer. Why are the genes in the first place not associated with the super-enhancer based on the RNA-seq and ATAC-seq data, instead of taking

the gene with closest TSS? After this initial annotation, a further prioritization can be performed based on the correlation between the H3K27ac ChIP and RNA-seq data. The way the analysis is performed here probably results substantial misannotated SE-gene associations and thus loss of data.

We agree, SE-gene associations invariably involve loss of data and false negatives. We did, however, include further prioritization based on correlation between H3K27ac ChIP-seq and RNA-seq expression data. While this may have missed some, this filtering did prevent inclusion of genes that were associated by TSS proximity but which were anti-correlated with H3K27ac changes.

12. Please clarify where the “lost super-enhancer” associated with SOX11 is located. There is discrepancy between Fig 3D and Fig 5A. The coordinates of the SOX11 downstream super-enhancer in Fig 3D are chr2:6427620-6453786. However, in Fig 5A and supplementary Fig 5A, the “lost super-enhancer” associated with SOX11 is marked more upstream. In Fig 5A, the super-enhancer defined in Fig 3D can be found where 2 orange stripes are next to each other. And according to the HiC data, the super-enhancer shown in Fig 3D is as such not interacting with the SOX11 locus. Why is the focus in Fig 3D not on the lost SE?

Fig 3D shows 2 lost SEs, and SOX11 has a total of 4 lost SEs. Upon further inspection of the HiC data, we have (1) numbered and named these SEs to make the connection between Fig 3D and 5A more clear and (2) we have found that in the HiC data, there is connectivity among all of the SEs in the lost cluster near SOX11, and have updated Fig 5 (new Fig. 6a) accordingly.

In addition, while in Supplementary Fig 5A, a clear reduction in K27ac is observed, this is not the case for the ATAC-seq profiles. Please explain. Is there a possibility that the SE lost its activity (lost H3K27Ac) but chromatin is still open?

We agree with the reviewer and indeed think that despite loss in H3K27ac levels at SOX11 SE, that chromatin may still remain open after 4Days ATRA treatment. This also is also supported by our SOX11 cut and run experiments after 4D ATRA treatment, where we observed minimal changes in SOX11 binding despite losing its SEs. However, at many SOX11 bound genes that showed no decrease in SOX11 binding, we still observed decreased gene expression (n = 182 genes) after ATRA treatment.

13. Fig 1C: Read depth can influence the position of the identified super-enhancers along the Y-axis, as higher read depth will result in increased H3K27ac signal. Please comment on read depth for every condition or possible normalisation strategies you used.-

Normalization process for read depth has now been described in methods sections.

14. Fig 5D: The effect on SOX11 mRNA expression is limited. Possible several SEs need to be targeted rather than in order to achieve full reduction of SOX11 mRNA levels? This should be commented and if possible tested?-

We agree with the reviewer regarding the necessity for the loss of multiple SEs for complete loss of SOX11, However, we believe these experiments are better suited for a more focused study on the role of SOX11.

16. Supplementary Fig 5C: Based on only 2 non-MYCN amplified cell lines it is difficult to conclude the claim that MYCN predisposes the cells to SOX11 dependence. In contrast, this low dependency is

probably more correlated to SOX11 expression. To address this, the dependency data could be correlated with SOX11 expression. In this context, it is important to note the CLBGA cell line has no MYCN overexpression but high SOX11 levels.

We agree that given NB cell line heterogeneity one cannot make blanket statements. In our original manuscript (now in revised manuscript) we stated there was a range of dependencies. The DepMapQ3 evaluating 32 NB cell lines indicate that almost 1/3 of NB cell lines tested show significant dependencies (-1.2 to -0.6). We have included a revised graph in our revised Suppl. Fig S8c, showing the relationship between SOX11 dependency and mRNA expression. We have now edited the text in the revised manuscript Results Section: SOX11 inhibition decreases cell growth in NB cells.

Minor:

- In general, more information is needed in the figure legends and at the figures: statistical test used (missing in Fig S2E, Fig 6D, Fig 6F-G), error bars and what they mean (missing in Fig S1B-C and S2E), number of replicates and if they are biological or technical (missing in Fig. S1A-C), timepoint of read-out (missing in Fig 6C and D, Fig S5D), normalisation strategy (Fig 2F), statistics (missing for Fig 4C LAN5 and Fig 4I) - Legend Fig S1A: the word “decrease” is 2 times used in the sentence, please correct.

We thank the reviewer for pointing out the inadvertent mistake. We have now corrected these in the Figure legends.

- Fig S1D: In the text, there is a reference to the genes MYCN, ISL1 and GATA3, but in the heatmap only MYCN and ASCL1 are depicted for the downregulated genes.

We have now called out those additional genes in the heatmap.

- Fig S2D: Indicate in figure legend that this is about TF-associated SEs. At this moment it is not clear what the difference is between Fig 1D and S2D.

The TF associated SEs in Figure S2D (now Figure S3D) are a subset of SEs in Fig 1D, those associated with TFs only.

- This sentence is difficult to read: “RNA-seq indicated changes in SEs signal of these TF associated SEs (Suppl. Fig. S2d) corroborated changes in the downstream gene expression (Suppl. Fig. S2e).” We have changed this sentence to read: “The changes in SEs signal of these TF associated SEs (Suppl. Fig. S2d) is supported by the changes in downstream gene expression (Suppl. Fig. S2e).”

- In the manuscript, there is no reference to Supplementary Figure 3.

Supplementary Figure 3 which is now revised Supplementary Figure 4, is referred in Results section: Super-enhancers fluctuate in sequential waves, Page 9.

- Please check annotation of the references to supplementary figures and be consistent: “Suppl” vs “Sup”.

- Some genes needed to be in italics are not.

- Figure legend Fig S5D concerns NTRK1, but NTRK1 is not marked in the figure. Actually, the same sentence is used in the legend as for Fig S5G...

- Concerning the sentence hereafter: “However, silencing of NB CRC components GATA3, PHOX2B, ISL2 and TBX2 (Suppl. Fig. 5g) did not lead to decreases in SOX11 (Fig. 6g)”. Why is there a reference to Suppl. Fig. 5g which shows NTRK1 expression upon SOX11 knockdown?

We thank the reviewer for pointing out the inadvertent mistakes. We have now corrected them in the revised manuscript.

Reviewer #3 (Remarks to the Author):

This is a well written, interesting manuscript by Banerjee et al. focused on the transcriptional reprogramming that underlies the differentiation of neuroblastoma (NBL) cells with retinoic acid. Retinoic acid is a standard treatment for children with high-risk neuroblastoma, and there has been a renewed interest in differentiation therapy in cancer with the approval of drugs such as IDH inhibitors in leukemia. The work is timely and important.

The authors study the effects of ATRA in a time course in NBL cell lines by RNA sequencing and ChIP sequencing. They focus on the role of SOX11 as an oncogenic driver in NBL and the activation of SOX4 as important in the differentiation process.

All told, this is a strong manuscript although there are some questions to address before publication.

Figure 1. The authors state that there is no change in the enhancer landscape in the EtOH controls over the time course compared to the ATRA treated cells, but they do not show the data. It would be important to show an overview of this data in the supplement, including the controls, to evaluate clustering of the samples (e.g., t-SNE plots in the space of TE or SE, which includes controls and ATRA treated samples).

We thank the reviewer for the suggestion. As mentioned in the Results under subheading “All-trans retinoic acid (ATRA) mediated differentiation leads to dynamic changes in super-enhancer landscape of MYCN-amplified NB cells”, the enhancer landscape wrt total number of TE or SE identified in controls essentially remain constant, however, there are SEs whose temporal pattern changed in both EtOH and ATRA treated cells. We sub grouped those as SEs with Max variance <200 H3K27Ac, n=116. These SEs along with SEs that remained stable over the course of treatment (SD < 30%; n=143), were removed prior to clustering analysis, helping us to focus only on the SEs dynamically regulated by ATRA. For the remaining SEs (n=940), K-means clustering based on H3K27ac density z-score was used and identified 4 temporal patterns of regulated SEs (Suppl. Fig. S1i). However, according to the suggestion we have now provided PCA plots for the SE identified in controls and ATRA treated samples, in the new Suppl. Fig. S1j:

There is a decrease in the SE at MYCN at 2 and 4 hours but it looks like it is regained in the 3rd wave. With chronic ATRA treatment, does MYCN increase at time points after 8 days? What happens to MYC in terms of expression or based upon SE/TE binding? Does MYC compensate for MYCN?

We thank the reviewer for raising this point. This concern was raised by another reviewer, and we found MYCN SEs were not perfectly labelled in the rank-ordered graph and have now fixed the issue. To answer the question, by proximity analysis, MYCN is driven by 3 different SEs, indeed the SE signals is regained in 3rd wave compared to Day 2 and Day4, as shown in the table above. However, the H3K27ac levels of all 3 SE at Day 8 still remains 1.4 to 5.4 fold below the levels in control cells. MYCN is not expressed at Day 8. As seen in figure 2d, MYCN mRNA levels remain decreased at Day 8. In addition, MYCN protein levels stays low in differentiated cells at Day 8 as observed in Suppl. Fig. S2f.

The NB cells used in the study KNCR and LAN5 are MYCN amplified and have minimal to no expression of MYC. Our previous studies in the laboratory have shown no change in MYC levels with ATRA treatment, and do not compensate for MYCN expression.

What are the motifs enriched in the subset of enhancers with a decrease in H3K27Ac binding over the time course?

Using HOMER Motif finder, we found that in sites of decreased H3K27ac binding (focusing on nucleosome free regions between H3K27ac peaks), the motifs in lost sites were enriched in the ATOH1/NeuroD/ASCL1 bHLH (motif: CAGCTG), the GATA Zf motif (GATAAG), and the LHX2/ISL1 homeobox TAATT(A/G).

Figure 2. How do the authors interpret the western blot in S2f for targets such as GATA2 and GATA3 where there is a pretty striking change in expression of these genes in the EtOH treated cells over time? For example, while GATA3 is decreased compared to EtOH at each time point, GATA3 is more highly expressed even in the ATRA treated cells at 8D compared to control at 2D.

We agree with the reviewer that GATA3 and GATA2 expression change on the control samples over time. For GATA3 looking at the SEs driving GATA3 expression, in addition to the Lost SEs controlling GATA3 expression, an additional SE in the Second wave cluster is linked with GATA3 expression. We hypothesized that after the initial downregulation of GATA3 after 2D and 4D ATRA treatment, the activation of the Second Wave cluster SE at 4D leads to resurgence of GATA3 expression at 8D to a level higher than the 2D control. This is also a part of the limitation of our assay system, where there is potential activation of the ATRA resistance population, that limits our capability to make any assessment of the changes in the landscape at or beyond this time point. We have now included a section regarding the limitation of our system in the discussion section. (I don't see this part in discussion section)

A table should be included showing the genes that are "validated" in the LAN5 cells that have similar changes in expression as in the KCNR cells with ATRA treatment in each cluster (Fig. 2e). We thank the reviewer for the suggestion. We have now included the list of genes that are validated in LAN5 cells in Suppl. Fig.S4c.

Since the focus of the paper is on SOX genes, the authors might comment in the discussion on their result that SOX21 also seems to be repressed in the lost cluster.

We have inserted a comment in the discussion and a reference. Edited text "In normal murine cells Sox21 is required for Sox2 induced reprogramming of fibroblasts. Our finding of the loss of the SOX21 SE and decreases in SOX21 mRNA expression suggest an involvement in NB cell

growth although more detailed future studies are needed to examine the role of SOX21 in NB more fully.”

Figure 3. *In Fig 3d, it is notable that the NBL models labeled adrenergic tend to have increased H3K27Ac load for SOX11 compared to the more mesenchymal cell lines, while the opposite is true for SOX4. The authors should comment on this. Are the adrenergic/mesenchymal signatures from previously published work (Von Groningen et al) enriched in the ATRA treated cells compared to the vehicle treated cells?*

We appreciate the reviewer’s comment. Our clustering analysis of NB cells utilizing published H3K27ac ChIP-seq data from Von Groningen et al, 2017, Nat Genet, and H3K27ac data from KCNR cells, shows KCNR cell line are adrenergic in nature. Both control cells and ATRA treated cells cluster along with other ADRN cell lines based on published literature. In addition, the expression of SOX11, according to Von Groningen et al analysis is indeed higher in adrenergic cell lines compared to mesenchymal cell line (Von Groningen et al, 2017, Figure 3). This is consistent with increased H3K27ac load for SOX11 in ADRN cells compared to MES cell lines.

In the survival plots for SOX11, does SOX11 expression track with MYCN? Is this significant if corrected for MYCN or stage? –

We find that the SOX11 expression correlates with MYCN expression. Yes, SOX11 survival plots remain significant even if only Stage 3 and 4 NB patients are analyzed. However in the MYCN amplification subset of patients, SOX11 does not have any associated with survival. We have now included the stage corrected survival plots in new Suppl. Fig. S6c and mentioned it in the main text.

Figures 4 and 5 are strong with clear data and interpretations. Is SOX4 Overexpression sufficient to induce differentiation? This experiment would further strengthen the work.-

We agree with the reviewer and have performed extensive studies to evaluate the effects of over-expression of SOX4 in KCNR cells. We have found that over-expression of SOX4 alone has no effect on cell growth, morphologic differentiation or selected differentiation associated genes. The referenced figures supporting these statements are found in Suppl. Fig. S7a-c. A statement in the Discussion notes this as well.

Figure6. *The authors seek to place SOX4 and SOX11 as members of a CRC in NBL. In this case, they need more supportive data. They could use publically available ChIPSeq data to demonstrate that other CRC members bind to enhancers for SOX11 and potentially SOX4 in NBL. This analysis should be done. Similarly, to definitively show Figure 6h, it would be important to demonstrate that SOX11 and SOX4 bind at enhancers for other CRC members. This reviewer recognizes the challenges of ChIP-seq when commercial antibodies are not effective. While potentially beyond the scope of the current manuscript, the authors could attempt ChIPseq with HA-tagged versions of the proteins, particularly for SOX11, or maybe rather tone down the claims that SOX11 sits atop the CRC hierarchy.*

To clarify we would not place SOX11 in the CRC but rather our data would indicate that it is more an input to the CRC since inhibition of established/published CRC members does not lead to changes in SOX11 expression. We also would not place SOX4 in the NB CRC controlling self-renewal but rather would put it in a more differentiated CRC as denoted in new Fig. 7h.

To address whether SOX11 or SOX4 bind to CRC members, we performed SOX4 and SOX11 CUTand RUN (analogous to ChIP-seq) in KCNR cells treated with or without ATRA for 4 days. Commercial antibodies for SOX11 were not optimal but available ChIP grade antibodies for SOX11 showed binding to CRC members but little overall change after ATRA treatment. SOX4 showed little binding to CRC members under control conditions but showed increases in selected CRC members after ATRA treatment

(Suppl. Fig. S9). Genome wide we saw clear increases in SOX4 binding after ATRA treatment. These new SOX4 sites were strongly enriched in the SOX4 motif and were found near genes which on average an increase in transcriptional output (new Fig 5). Importantly, SOX4 showed binding to the differentiated CRC TFs RARA, PPARG, NFIA and GATA2 (Suppl. Fig. 9b).

Are the SOX11 siRNA gene expression changes enriched for the ATRA signature by GSEA or some other test for statistical enrichment?

The Frumm differentiation gene signature (based on an ATRA differentiation signal, Frumm et al PMID:23706636) was not enriched after SOX11 siRNA knockdown. However, as noted in Suppl. Fig. S8f, after silencing of SOX11 there were enrichments in genes associated with nervous system development, growth and branching of neurites (Suppl. Fig. S8f). Moreover there were increases in NTRK1 and increased responsiveness to NGF, a hallmark of sympathetic neural cells after SOX11 silencing as noted in Suppl. Fig. S8g.

The authors might also comment that loss of HAND2 seems to increase SOX11 expression similar to the pattern seen for MYCN.

We have mentioned that loss of HAND2 seems to increase SOX11 expression in the results section and we have referenced this study (PMID:36598365).

REVIEWERS' COMMENTS

Reviewer #2 (Remarks to the Author):

The authors have carefully addressed the proposed comments and replied accordingly and added additional experimental data where needed.

Reviewer #3 (Remarks to the Author):

The manuscript revision by Banerjee et al., focused on the temporal regulation of super-enhancer programs during retinoid differentiation of neuroblastoma cells, is much improved from the original submission. The authors have adequately addressed my concerns/questions, and I would be happy to see this manuscript now published in Nature Communications.

There is just one small issue to resolve - on p. 20 of the response to the reviews there is an internal comment still in red that I believe the authors forgot to address before the submission.